# The global ocean mixed layer depth derived from an energy approach based on buoyancy work

Efraín Moreles[1], Emmanuel Romero[2], Karina Ramos-Musalem[3], and Leonardo Tenorio-Fernandez[4,5]

[1]Instituto de Ciencias del Mar y Limnología, Universidad Nacional Autónoma de México, Mexico City, Mexico
[2]Departamento Académico de Sistemas Computacionales, Universidad Autónoma de Baja California Sur, Carretera al sur km 55, Col. Mezquitito, La Paz, BCS, Mexico
[3]Department of Physical Oceanography, Centro de Investigación Científica y de Educación Superior de Ensenada, Ensenada, Baja California, Mexico
[4]Instituto Politécnico Nacional-Centro Interdisciplinario de Ciencias Marinas, Departamento de Oceanología. Av. Politécnico Nacional s/n. Col. Palo de Santa Rita, 23096, La Paz, Baja California Sur, Mexico
[5]CONHACyT, Consejo Nacional de Humanidades, Ciencia y Tecnología. Av. Insurgentes Sur 1582, Col. Crédito Constructor, Demarcación Territorial Benito Juárez, 03940, Mexico City, Mexico

**Correspondence:** Efraín Moreles (moreles@cmarl.unam.mx)

**Abstract.**

The mixed layer depth (MLD) is critical for understanding ocean-atmosphere interactions and internal ocean dynamics. Traditional methods for determining the MLD, commonly relying on constant temperature and density thresholds, may not adequately address spatial and temporal variations in local oceanographic conditions, thereby limiting their global consistency and applicability. An energy-based definition of the mixed layer could be a more physically consistent alternative to address this issue. We propose a physically derived and energy-based methodology that defines the mixed layer as the energetically homogeneous upper ocean layer in which water parcels can move with little or no buoyancy work. The threshold in buoyancy work, which determines the mixed layer globally throughout the year, was carefully investigated. An energy-based global monthly MLD climatology demonstrated the reliability of the methodology across diverse ocean conditions and its usefulness for studies spanning seasonal to climate time scales, from regional to large spatial scales. Our easy-to-implement MLD methodology provides a robust criterion that is globally and temporally consistent, maintaining quasi-homogeneity in energy, density, and temperature year-round for most of the global ocean. This study promotes the development of MLD energy-based methodologies that could offer significant potential for advancing the study of dynamic and thermodynamic processes, including heat content and vertical exchanges. Our methodology could also serve as a robust tool for validating ocean circulation models and supporting intercomparison studies in initiatives such as the Ocean Model Intercomparison Project (OMIP) and the International Coupled Model Intercomparison Project (CMIP). Future research will explore its applicability to high-frequency processes and regional variability, further enhancing its utility for understanding and modeling oceanic phenomena.

# 1 Introduction

The ocean mixed layer is the ocean's surface layer in direct contact with the atmosphere, whose properties (potential density, temperature, salinity, and other tracers) are relatively homogeneous in the vertical. Such relatively vertical homogeneity is due to turbulence caused by wind effects, buoyancy-driven fluxes, and waves (D'Asaro, 2014; Sallée et al., 2021; Reichl et al., 2022). The mixed layer concept allows for diagnosing vertical exchanges within the ocean and those between the ocean and the atmosphere without a detailed analysis of the associated turbulent processes (D'Asaro, 2014; Sutherland et al., 2014;

Franks, 2014). The mixed layer plays a crucial role in the Earth's weather and climate since it determines the energy, mass, and momentum exchanges between the ocean and the atmosphere (Gill, 1982). It largely determines different aspects of the climate system: ocean surface temperature (Deser et al., 2010), formation and properties of water masses (Hanawa and D.Talley, 2001; Groeskamp et al., 2019), thermal energy available to a tropical cyclone (Shen and Ginis, 2003), biological productivity (Franks, 2014; Bouman et al., 2020), chlorophyll content (Briseño Avena et al., 2020; Carvalho et al., 2017), and carbon subduction

(Bopp et al., 2015; Omand et al., 2015).

The mixed layer depth (MLD) is a key variable in understanding the past, present, and future variability of Earth's weather and climate (Sallée et al., 2021; Treguier et al., 2023). However, the definition of the mixed layer as a relatively homogeneous layer is very vague (de Boyer Montégut et al., 2004), which has led to numerous definitions and estimates of the MLD (de Boyer Montégut et al., 2004; Holte and Talley, 2009; Schofield et al., 2015; Reichl et al., 2022; Romero et al., 2023). The

above has resulted in high uncertainty in estimating the MLD, mainly in deep and intermediate water formation regions, polar seas, and barrier layer regions (Treguier et al., 2023). That also has affected the analysis of the relationship between the MLD and various ecological processes, such as chlorophyll-$a$ content, phytoplankton dynamics, and primary production (Carvalho et al., 2017; Bouman et al., 2020). No MLD definition provides accurate estimates for all world regions under different ocean conditions throughout the year.

Different authors have suggested that the physically relevant definitions of MLD should be density-based since this approach captures both temperature- and salinity-driven stratifications at the mixed layer (Griffies et al., 2016; Sallée et al., 2021; Treguier et al., 2023). The protocol used to compute the MLD in the Ocean Model Intercomparison Project (OMIP) and the International Coupled Model Intercomparison Project (CMIP) considers a constant density threshold; however, in regions with vertically compensated layers, the density threshold may overestimate the MLD (de Boyer Montégut et al., 2004). No physical reason

sustains the choice of a specific density threshold; instead, it is heuristically obtained. Reichl et al. (2022) proposed an MLD definition considering the potential energy anomaly of the water column; their work is promising because it is based on physical principles, but they did not provide specific energy values to define the MLD globally during all seasons. Consequently, MLD methodologies, physically derived and energy-based, need to be developed and investigated in more detail to provide accurate estimates for all world regions under different ocean conditions (Treguier et al., 2023).

This study aims to develop a physics-based methodology, based on energy considerations, for calculating the MLD under different ocean conditions. The mixed layer is defined from an energy measure of the vertical homogeneity of the water column, which quantifies the work done by the buoyancy force to displace a water parcel vertically. The methodology's global appli-

cability is one of its crucial contributions; therefore, long-term observational data providing extensive global coverage were used to construct a gridded MLD climatology. The resulting global monthly MLD climatology and the value of the buoyancy work determining the mixed layer during each month were carefully investigated and compared with other MLD methodologies. The energy-based definition of the MLD is consistent with physics; it has the potential to provide further insights into various dynamic (e.g., vertical exchanges within the ocean and between the ocean and the atmosphere), thermodynamic (upper ocean heat content), and ecological (e.g., chlorophyll-$a$ content and phytoplankton dynamics) processes. The observation-based global MLD climatology could be a reference to validate numerical solutions, perform MLD model intercomparison studies, and provide new insights into understanding the mixed layer.

## 2   Methodology and data

### 2.1   An energy measure of the vertical homogeneity of the water column

Previous research has established the mixed layer as the upper ocean layer that is relatively homogeneous in the vertical. Several approaches to quantify the upper ocean layer's vertical homogeneity have been proposed from this premise. This study proposes a quantitative measure of the vertical homogeneity of the upper ocean layer, derived from physical principles and based on energy considerations, from which the MLD can be defined. Since vertical changes in density hinder turbulence and subsequent mixing, a physically relevant metric of the water column's vertical homogeneity should be density-based. Thus, we propose quantifying the water column's vertical homogeneity in terms of the work done by the buoyancy force in vertically displacing a water parcel under static instability conditions, herein referred to as the work done by buoyancy (WB). When considering vertical displacements from the ocean's interior to the surface, WB can constitute a proxy for the vertical homogeneity of the water column: the lower the WB, the greater the vertical homogeneity of the water column, and vice versa.

The following shows the mathematical development to define WB. Consider a fluid in hydrostatic balance in which a water parcel is adiabatically displaced along the vertical; in such a displacement, the potential density of the parcel is materially conserved. The buoyancy force experienced by a parcel, initially at equilibrium at $z_{\mathrm{eq}}$, when displaced from $z_{\mathrm{eq}}$ to any depth $z$ is given by (Vallis, 2006, p. 92)

$$F(z) = g \left[ \rho^\theta(z) - \rho^\theta(z_{\mathrm{eq}}) \right], \tag{1}$$

where $g$ is the acceleration due to gravity and $\rho^\theta$ is the potential density of the environment referred to the pressure at level $z$. The force is null at $z_{\mathrm{eq}}$, positive if $\rho^\theta(z) > \rho^\theta(z_{\mathrm{eq}})$ and negative if $\rho^\theta(z) < \rho^\theta(z_{\mathrm{eq}})$. By calculating the line integral of the buoyancy force along such a displacement, WB is obtained,

$$\mathrm{WB}_{z_{\mathrm{eq}} \to z} = \int_{z_{\mathrm{eq}}}^{z} F(\gamma) d\gamma = g \int_{z_{\mathrm{eq}}}^{z} \left[ \rho^\theta(\gamma) - \rho^\theta(z_{\mathrm{eq}}) \right] d\gamma = -g \left( z - z_{\mathrm{eq}} \right) \rho^\theta(z_{\mathrm{eq}}) + g \int_{z_{\mathrm{eq}}}^{z} \rho^\theta(\gamma) d\gamma. \tag{2}$$

Since we are interested in determining the ocean mixed layer, $\rho^\theta$ is referred to the surface (to 0 dbar), $\rho_0^\theta$. The limits in the integral of Eq. (2) typically should extend from the MLD to the free surface. Recent studies have utilized the integral $\int_{\text{MLD}}^0 \left[ \rho^\theta(z) - \rho^\theta(\text{MLD}) \right] dz$, commonly referred to as the vertically integrated stratification relative to the MLD, to examine the MLD and mixing in the Indian and Southern Oceans (Lee et al., 2011; Small et al., 2021; Caneill et al., 2024). WB depends on the cumulative effect along the vertical of the buoyancy force, which in turn depends on the difference in $\rho^\theta$ between any depth $z$ and $z_{\text{eq}}$. In stable density profiles, where $\rho^\theta$ increases with depth, WB is negative for upward and downward displacements of the water parcel; the displaced water parcel tends to return to its original depth $z_{\text{eq}}$ where it was at equilibrium. For an upward displacement (positive displacement), the force is downward; for a downward displacement (negative displacement), the force is upward. The opposite behavior is obtained in unstable density profiles, where $\rho^\theta$ decreases with depth.

WB quantifies the energy required for a water parcel to displace vertically from its equilibrium level to any level; it represents the potential energy barrier to its displacement. Consequently, WB is better than density for diagnosing the water column section in direct contact with the atmosphere and its vertical homogeneity in energetic terms. Moreover, WB yields an estimate of the buoyancy flux required to mix an upper water column. The buoyancy flux is a major driver of vertical processes; it determines the depth of vertical exchanges within the ocean and, consequently, the MLD (Gill, 1982; Sutherland et al., 2013; Zippel et al., 2022). The relationship between the buoyancy flux and WB is evinced by showing that it corresponds with columnar buoyancy, which represents the buoyancy loss required to mix the water column from the surface down to a depth of $h$ during a specific time period (Lascaratos and Nittis, 1998; Herrmann et al., 2008); dividing columnar buoyancy by this time period gives the buoyancy flux required to mix that column during the same time period (Faure and Kawai, 2015). To show the above, we employed an approximation for columnar buoyancy in terms of $\rho_0^\theta$, valid for shallow depths, derived by Caneill et al. (2024),

$$-\int_h^0 z N^2(z) dz \simeq -\frac{g}{\rho_0} \int_h^0 \left[ \rho_0^\theta(z) - \rho_0^\theta(h) \right] dz, \tag{3}$$

where the left-hand side is columnar buoyancy, $h$ is the depth at which columnar buoyancy is calculated, $N^2$ is the square of the buoyancy frequency, and $\rho_0$ is a density of reference. Caneill et al. (2024) employed Eq. (3) to calculate columnar buoyancy at 250 m depth. Figure S1 in the Supplement shows an example of the differences in columnar buoyancy obtained using $\rho_0^\theta$ and the locally referenced potential density for a location in the Greenland Sea during April. The mean absolute percentage error is negligible in shallow depths (3.2% in the first 300 m of the water column), small at medium depths (14.3% in the first 900 m of the water column), and significant at very great depths (32.1% in the first 1800 m of the water column). Even in this deep-water formation region, with strong convective processes, the differences between both expressions of columnar buoyancy are minor in the first few hundred meters. The above adds confidence in using the approach given by Eq. (3) in the context of our study.

From Eq. (3), WB can be expressed as

$$\text{WB}_{h \to 0} = g \int_h^0 \left[ \rho_0^\theta(z) - \rho_0^\theta(h) \right] dz \simeq \rho_0 \int_h^0 z N^2(z) dz. \tag{4}$$

Equation (4) shows that WB approximately corresponds with columnar buoyancy in shallow depths; WB is the negative of columnar buoyancy multiplied by $\rho_0$, and consequently, yields an estimate of the buoyancy flux. Since the buoyancy flux is a term in the turbulent kinetic energy budget of the upper ocean layer (Zippel et al., 2022), it is plausible to connect WB with this energy budget and, thus, with the physics of boundary layer turbulence. The above suggests that our methodology for defining the MLD is consistent with the turbulent approach to the mixed layer formation; however, as described below, our mixed layer definition does not account for the timescales of active mixing, somehow blurring the connection between WB and the turbulent formation of the mixed layer. Further analysis of this connection is beyond the scope of this study and is proposed for future research.

The use of WB as a proxy for the vertical homogeneity of the water column can be exemplified by considering typical potential density profiles in different seasons (Fig. 1). The WB required to displace each water parcel in the water column from its equilibrium level to the surface is also shown for each density profile; to better appreciate the relationship between density and WB, the negative value of WB is plotted in all the figures. A strong correspondence exists between the density profile and its associated WB profile: a perfectly homogeneous water column has zero WB values, whereas a stratified water column has density and WB increasing with depth. However, due to the nonlinearity between WB and density, the density variation between two depths is not proportional to the corresponding variation in WB. For example, the stratified profile (Fig. 1d) has a larger density variation than the winter profile (Fig. 1b), but the WB variation is larger in the winter than in the stratified profile: large density variations do not always correspond to large WB values.

## 2.2 Defining the mixed layer

The mixed layer definition taken in this study is based on that of Brainerd and Gregg (1995), who defined it as the zone in which surface fluxes have been mixed through timescales longer than several days. The mixed layer does not address the diurnal mixing cycle or the timescales in which mixing is currently active, that is, the timescales relevant to the mixing layer. The mixed layer considered in this study is representative of seasonal timescales; it is the relatively homogeneous upper ocean layer formed by the history of mixing when the ocean is nearly in thermal equilibrium with the atmosphere on timescales of a few days or more (Brainerd and Gregg, 1995). This study defines the mixed layer as the energetically homogeneous upper ocean layer; we consider a layer to be energetically homogeneous when water parcels can move with little or no WB within it. This energetic homogeneity is the key and distinctive property of our mixed-layer definition. Since WB is a proxy for the vertical homogeneity of the water column, the mixed layer is thus the upper ocean layer with small WB values, which is well-mixed in energetic terms and, therefore, in contact with the atmosphere.

How small should the WB values be to characterize a well-mixed layer? To answer this question, we will continue the mathematical development of WB. From Eq. (2), when considering the vertical section from any depth $h$ to the free surface $\eta$ and using the first mean value theorem of calculus for definite integrals, we can express Eq. (2) as a relationship between the degree of density inhomogeneity $\Delta \overline{\rho_0^\theta}$ of any upper section of the water column and the corresponding WB required to displace

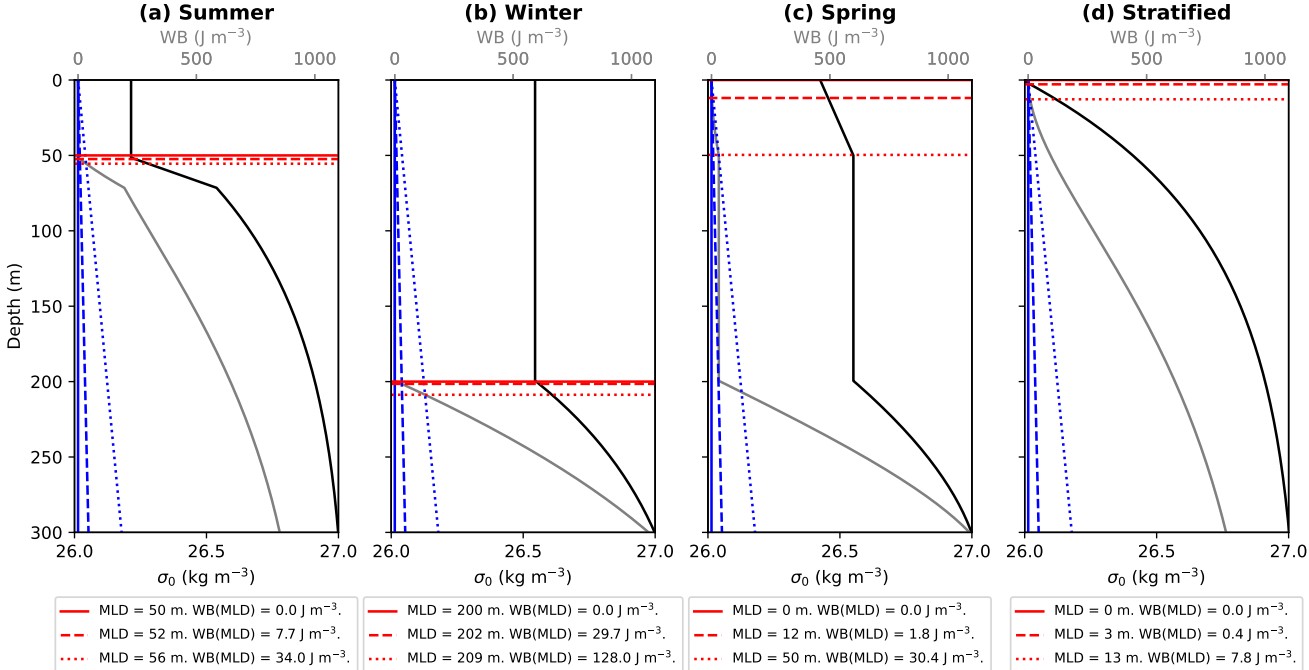

**Figure 1.** Typical density profiles for (a) summer, (b) winter, (c) spring, and (d) a stratified condition (black curves) and the corresponding WB (gray curves). The density profiles, from Treguier et al. (2023) in their Fig. 2, show the potential density anomaly referred to 0 dbar ($\sigma_0$). Three $\Delta\overline{\rho_0^\theta}$ values (0, 0.0150, and 0.0625 kg m$^{-3}$) were considered; the larger the $\Delta\overline{\rho_0^\theta}$, the larger the slope of $\mathrm{WB_{ref}}(z)$ (blue curves). For each $\Delta\overline{\rho_0^\theta}$, the corresponding MLD (red curves) and WB value at the MLD are also shown.

a water parcel from its base to its top,

$$\mathrm{WB}_{h\to\eta} = -g(\eta - h)\left[\rho_0^\theta(h) - \overline{\rho_0^\theta}\right] = -g(\eta - h)\Delta\overline{\rho_0^\theta}, \tag{5}$$

where

$$\overline{\rho_0^\theta} \equiv \frac{1}{\eta - h}\int_h^\eta \rho_0^\theta(z)dz \quad \text{and} \quad \Delta\overline{\rho_0^\theta} \equiv \rho_0^\theta(h) - \overline{\rho_0^\theta}.$$

The intrinsic relationship between WB and the density variations along the water column can be explored via Eq. (5). We can explore the density structure of an energy-homogeneous layer and, reciprocally, the energy behavior of a density-homogeneous layer. Below, we describe these cases:

- For a layer with a unique, non-zero WB value, Eq. (5) establishes a nonlinear decrease of $\Delta\overline{\rho_0^\theta}$ with depth; the density variation should decrease with depth to maintain the same WB value (Fig. 2a). The degree of density inhomogeneity $\Delta\overline{\rho_0^\theta}$ along the water column changes according to the intensity of turbulence and mixing in the ocean: the greater the turbulence and mixing, the greater the vertical homogeneity of the water column (the density variation and the WB value

155   are small at large depths) and the larger the MLD. For example, $\text{WB} = 10\,\text{J}\,\text{m}^{-3}$ implies $\Delta\overline{\rho_0^\theta} = 0.015\,\text{kg}\,\text{m}^{-3}$ from the surface to 68 m depth and $\text{WB} = 20\,\text{J}\,\text{m}^{-3}$ implies $\Delta\overline{\rho_0^\theta} = 0.015\,\text{kg}\,\text{m}^{-3}$ from the surface to 136 m depth. The opposite behavior is obtained for low turbulence and mixing, which produces low vertical homogeneity of the water column (the density variation and the WB value are large at shallow depths) and a shallow MLD.

– A layer with a unique, non-zero $\Delta\overline{\rho_0^\theta}$ value is associated with a linear increase of WB with depth; more WB is required
160   to raise a parcel from a greater depth (Fig. 2b).

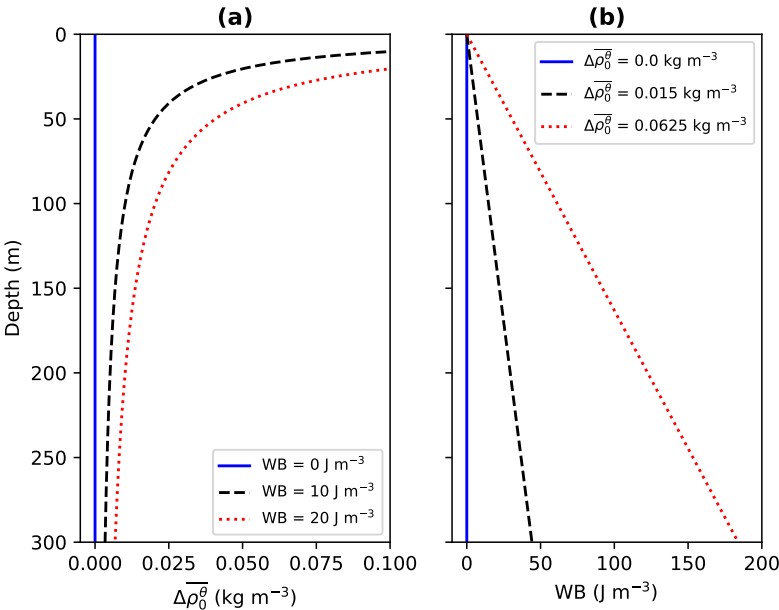

**Figure 2.** Relationship between the degree of density inhomogeneity ($\Delta\overline{\rho_0^\theta}$) along the water column and the corresponding WB required to displace a water parcel from any depth to the surface (Eq. 5). Profiles are shown for various single values of (a) WB and (b) $\Delta\overline{\rho_0^\theta}$.

Determining a well-mixed layer in energetic terms is not direct; finding the WB threshold requires a density-based reference value, i.e., the density variations along the mixed layer. An approach to finding the WB threshold that defines the MLD is to use Eq. (5), considering a specific degree of density inhomogeneity $\Delta\overline{\rho_0^\theta}$ along the mixed layer. The procedure is as follows:

1. Select a $\Delta\overline{\rho_0^\theta}$ value characteristic of a well-mixed layer in density; some of the density criteria suggested in the literature
165   can be used (Levitus, 1982; Kara et al., 2000; de Boyer Montégut et al., 2004). A $\Delta\overline{\rho_0^\theta}$ homogeneous in space and time will lead to spatially and temporally variable WB thresholds.

2. For the selected $\Delta\overline{\rho_0^\theta}$ value, use Eq. (5) to construct the associated reference curve of buoyancy work, $\text{WB}_{\text{ref}}(z) = -g(\eta - z)\Delta\overline{\rho_0^\theta}$. WB values smaller than $\text{WB}_{\text{ref}}$ indicate a layer quasi-homogeneous in density; in contrast, WB values larger than $\text{WB}_{\text{ref}}$ indicate a layer not quasi-homogeneous in density.

3. The intersection depth between the WB profile of interest and $\text{WB}_{\text{ref}}$ determines the vertical extension of the well-mixed layer in energetic terms for the profile of interest, that is, its MLD. The WB value at the MLD thus represents the WB threshold characterizing the well-mixed layer in energetic terms, according to the $\Delta\overline{\rho_0^\theta}$ value. Naturally, the WB threshold depends on the choice of the reference depth at which $\text{WB}_{\text{ref}} = 0$.

4. The resulting WB threshold should be analyzed to determine whether it identifies the entire vertical extent of the energetically homogeneous upper ocean layer with small WB values. If necessary, the WB threshold should be adjusted. The above determines its adequacy in producing a physically realistic MLD in energetic terms.

The above represents our approach to defining the MLD via a physics-derived, energy-based methodology, herein referred to as EBM. To differentiate our MLD methodology from other threshold-based methodologies, and for the sake of conciseness, we will refer to ours as an energy-based one; however, it is essential to emphasize that our MLD methodology is based on a specific energy approach, namely, the work done by buoyancy. Figure 1 also exemplifies the application of EBM in determining the MLD and the corresponding WB threshold, considering three $\Delta\overline{\rho_0^\theta}$ values. For idealized density profiles with a single strong pycnocline (summer and winter profiles in Fig. 1), the MLD is clearly defined (even by eye), but their MLD and corresponding WB threshold vary depending on the $\Delta\overline{\rho_0^\theta}$ considered. The MLD varies slightly (a few meters), but the WB threshold can vary significantly: it varies from 0 to 34 $\text{J}\,\text{m}^{-3}$ for the summer profile (Fig. 1a) and from 0 to 128 $\text{J}\,\text{m}^{-3}$ for the winter profile (Fig. 1b). For strongly stratified density profiles, profiles with several pycnoclines, or very smooth density profiles, the mixed layer can be shallow or deep depending on the chosen threshold characterizing quasi-homogeneous WB values. For the spring profile (Fig. 1c) with near-surface restratification, the mixed layer can be as shallow as 0 m, as deep as 50 m, or have intermediate depths depending on the WB threshold characterizing a quasi-homogeneous section. For the strongly stratified density profile (Fig. 1d), the mixed layer could be very shallow or non-existent. The variability in the WB threshold underscores the complexity of estimating the MLD.

The MLD definition differs between the potential energy (PE) anomaly methodology proposed by Reichl et al. (2022) and the methodology presented here. The PE anomaly diagnoses the MLD as the depth to which a given energy could homogenize a layer of seawater; the PE anomaly relates to the turbulent kinetic energy budget of the ocean surface boundary layer and serves as a good proxy for mixing, resulting in MLD estimates that are representative of active boundary layer turbulence. On the other hand, EBM determines the MLD by quantifying the vertical homogeneity of the water column in terms of the required WB to displace a water parcel vertically. WB is possibly connected to the turbulent kinetic energy budget of the upper ocean layer and, thus, to the turbulent approach of the mixed layer formation; however, we did not conduct an in-depth analysis of this connection, which we propose for future research.

## 2.3 Data

The Argo profiles for the global ocean (Wong et al., 2020) were used to compute the MLD. The profiles were obtained from the Argo snapshot of June 2024 and comprise data from January 2005 to December 2023 (Argo, 2024). Delayed mode profiles deemed good and probably good (quality flags 1 and 2) were selected, obtaining $\approx 2$ million in situ profiles. Using

the Thermodynamic Equation of SeaWater 2010 (McDougall and Barker, 2011), the conservative temperature ($\Theta$), absolute salinity ($S_A$), and surface-referenced potential density ($\rho_0^\theta$) were calculated for all the profiles, retaining their original vertical resolution. The spatial and temporal distribution of the Argo profiles used in this study is shown in Fig. S2 in the Supplement. The spatial coverage of the Argo data does not completely map the entire ocean (neritic and oceanic zones): coastal zones are consistently not mapped, and the data are somewhat scattered south of 60°S and scarce north of 60°N, mainly in the Pacific Ocean. Beyond these limitations, Argo data provide extensive global coverage and can be considered representative of the world ocean (Wong et al., 2020).

## 2.4 Construction of energy-based global monthly MLD climatologies

In this study, we constructed two energy-based global monthly MLD climatologies considering two $\Delta\overline{\rho_0^\theta}$ values, 0.0150 and 0.0625 $\mathrm{kg\,m^{-3}}$, characteristic of density variations of approximately 0.030 and 0.125 $\mathrm{kg\,m^{-3}}$ along the mixed layer, respectively (Levitus, 1982; Kara et al., 2000; de Boyer Montégut et al., 2004). The MLD was computed for each Argo profile within a 2°x2° grid cell for each month in the long-term record; the resulting MLD values were then averaged to obtain a representative value of the MLD for that grid cell and month. Since the original vertical resolution of the Argo profiles was retained, the resulting gridded MLD climatologies are observation-based. In determining the MLD, we used a reference depth of 10 m to avoid diurnal influences, in agreement with de Boyer Montégut et al. (2004) and Treguier et al. (2023). Therefore, we identified the energetically homogeneous layer below 10 m depth, considering the WB required to displace a water parcel from any depth to 10 m so that $\mathrm{WB}(10\,\mathrm{m}) = 0$. Potential density profiles were interpolated to 10 m if conservative temperature and absolute salinity measurements were not available at that depth; a linear interpolation of the potential density profile was implemented.

The characteristics of EBM were analyzed, and their performance was contrasted with three commonly used methodologies and a recent one, further expanding the applicability of this study. It would have been very significant to consider the PE anomaly in the comparison; however, it was not possible due to the lack of a specific criterion to determine the MLD on a global scale. The first two common MLD methodologies are the $0.03\,\mathrm{kg\,m^{-3}}$ and the 0.2°C thresholds proposed by de Boyer Montégut et al. (2004). The third common MLD methodology is the multi-criteria method of Holte and Talley (2009), which calculates possible MLDs derived from threshold and gradient methods to select a final MLD estimate based on physical features in the profile. The recent MLD methodology is the sigmoid function fitting method of Romero et al. (2023), which computes the MLD and the maximum thermocline depth by evaluating the fit of the sigmoid function to the temperature profile. We will refer to these methodologies as B04D, B04T, HT09, and R23, respectively (and collectively as the common MLD methodologies). All the MLD climatologies were calculated using the same Argo dataset, with the same temporal and spatial scales and reference depth, making them comparable.

## 3 Results

The results are presented in three parts. In the first part, we analyzed two energy-based global monthly MLD climatologies and the WB threshold that defines the mixed layer. We then evaluated the EBM performance using a quality index and explored the

degree of homogeneity in the mixed layer concerning density and temperature. In the second part, we compared EBM with the common MLD methodologies, considering their MLD magnitude and energy consistency in determining the MLD. The above represents an important contribution to previous studies on MLD climatologies, providing new insights into understanding the mixed layer. In the third part, we conducted a preliminary exploration of which WB values can define the MLD globally throughout the year.

## 3.1 Energy-based global monthly MLD climatology

The global monthly MLD climatology calculated with EBM, considering $\Delta\overline{\rho_0^\theta} = 0.0150\,\mathrm{kg\,m^{-3}}$, is shown in Fig. 3. In agreement with the expected physical behavior, the MLD exhibits a clear seasonality, being shallow during summer and deep during winter, and having a high heterogeneity in space. Figure 3 also shows the corresponding cumulative density function (CDF) of the MLD for the world ocean, considering all the months; thus, it represents the conjoint distribution in space and time of the MLD. Throughout the year, 50% of the world ocean has MLDs up to 44 m, while only 1% reaches MLDs over 269 m. The probability density function (PDF) used to compute the CDF was obtained through kernel density estimation (Rosenblatt, 1956; Parzen, 1962) using the gaussian-kde function from Python's SciPy library.

The following describes the MLD's spatio-temporal variability shown in Fig. 3. The tropical oceans have relatively shallow mixed layers throughout the year, with moderate seasonal changes; the MLD varies in a range of a few tens of meters. Semiannual cycles can be discerned in the region of barrier layer formation, approximately located in $[15°\mathrm{S}, 15°\mathrm{N}] \times [150°\mathrm{E}, 150°\mathrm{W}]$, and in the northern Indian Ocean, mainly in the Arabian Sea. In contrast to the tropical oceans, the regions from midlatitudes to high latitudes have deeper mixed layers with strong seasonal changes, with the MLD ranging from several tens of meters during summer and early fall to several hundred meters during winter and early spring. The seasonal changes are smaller in the North Pacific than in the North Atlantic and Southern Oceans. The MLD values in the North Pacific are asymmetric during wintertime; they are larger in the northwest than in the northeast. Concerning the seasonal behavior of the MLD in the Southern Ocean, the mixed layer is shallow in the continental shelves during each season, deepens in the Antarctic Circumpolar Current region, and becomes shallower towards the north of the Antarctic Circumpolar Current. The largest MLD values across the Southern Ocean are not located between the same latitudes; they are located between 60°S-50°S in the Pacific, between 50°S-40°S in the Indian Ocean, and between 60°S-50°S in the Atlantic Ocean. The largest MLD values occur during wintertime in deep and intermediate water formation regions and polar seas in the North Atlantic (south of Iceland and the Labrador, Greenland, Iceland, and Norway Seas) and in the South Pacific and South Indian Oceans between 65°S and 45°S. The MLD can reach values of up to 945 m in the Labrador Sea, 1074 m in the Greenland, Iceland and Norwegian seas region, and 614 m in the South Pacific and South Indian Oceans.

The global monthly climatology of the WB threshold characterizing the MLD, considering $\Delta\overline{\rho_0^\theta} = 0.0150\,\mathrm{kg\,m^{-3}}$, is shown in Fig. 4. This figure also shows the CDF of WB at the MLD for the world ocean, considering all the months. The spatial and temporal variability of the WB threshold is very similar to that of the associated MLD (Fig. 3), but with small variations in the WB magnitude through time. Equation (5) and Fig. 2 establish that the MLD derived from a unique, non-zero $\Delta\overline{\rho_0^\theta}$ threshold will not result in a unique, non-zero WB threshold. Interestingly, results for this $\Delta\overline{\rho_0^\theta}$ threshold showed that for most of the

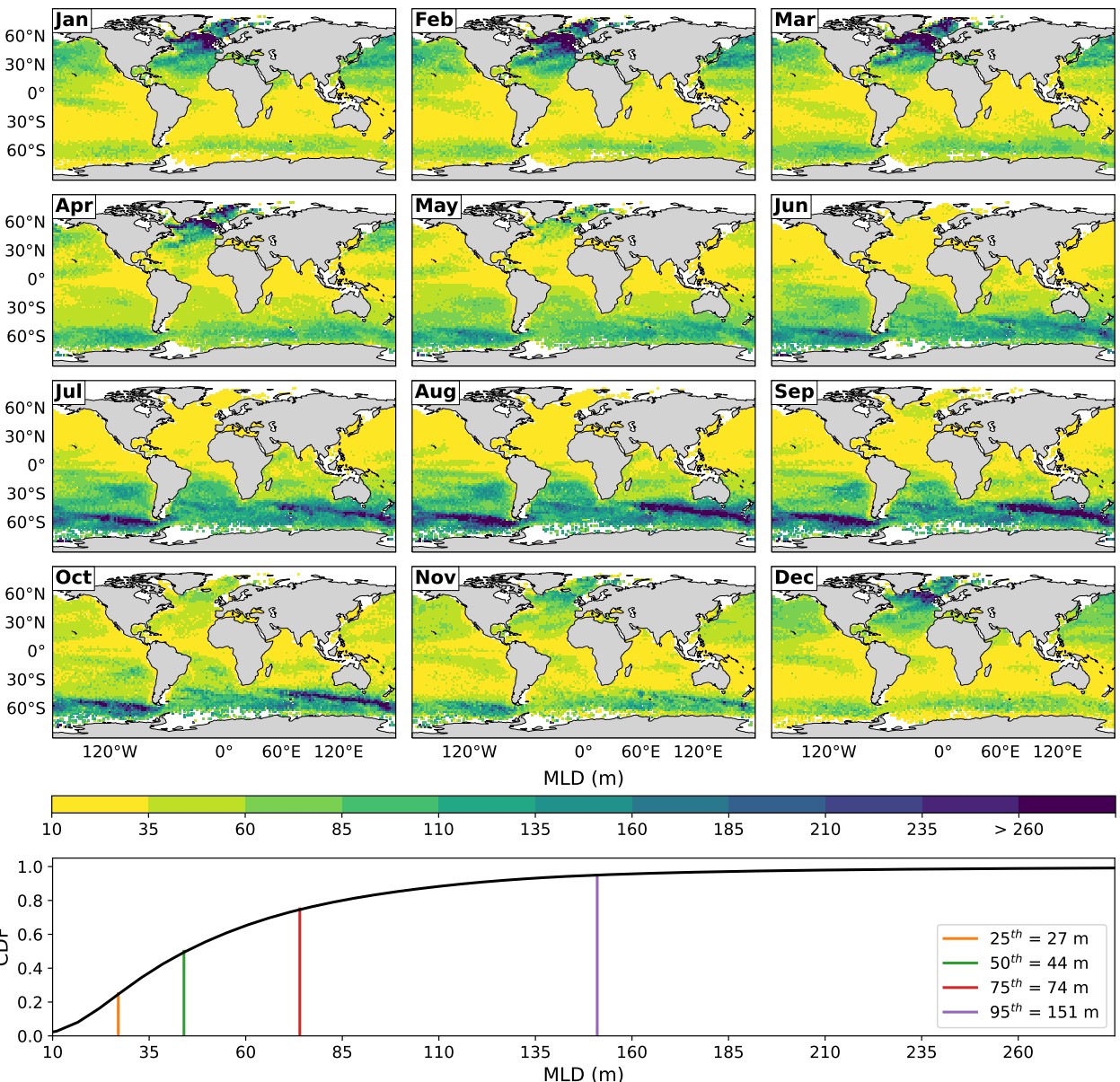

**Figure 3.** Upper panel: global monthly MLD climatology calculated with EBM, considering $\Delta\overline{\rho_0^\theta} = 0.0150 \ \mathrm{kg\,m^{-3}}$. Lower panel: the cumulative density function (CDF) of the conjoint distribution in space and time of MLD, with various percentiles shown. The maximum MLD is 1074 m.

world ocean, the WB thresholds seem small enough to characterize an energetically homogeneous ocean layer, which would be consistent with our energy definition of the mixed layer. 75% of the world ocean has WB thresholds not exceeding 9.5 J m$^{-3}$

year-round, and up to 95% has WB thresholds below $20.8\,\mathrm{J\,m^{-3}}$; the largest WB thresholds only occur in high latitudes during wintertime.

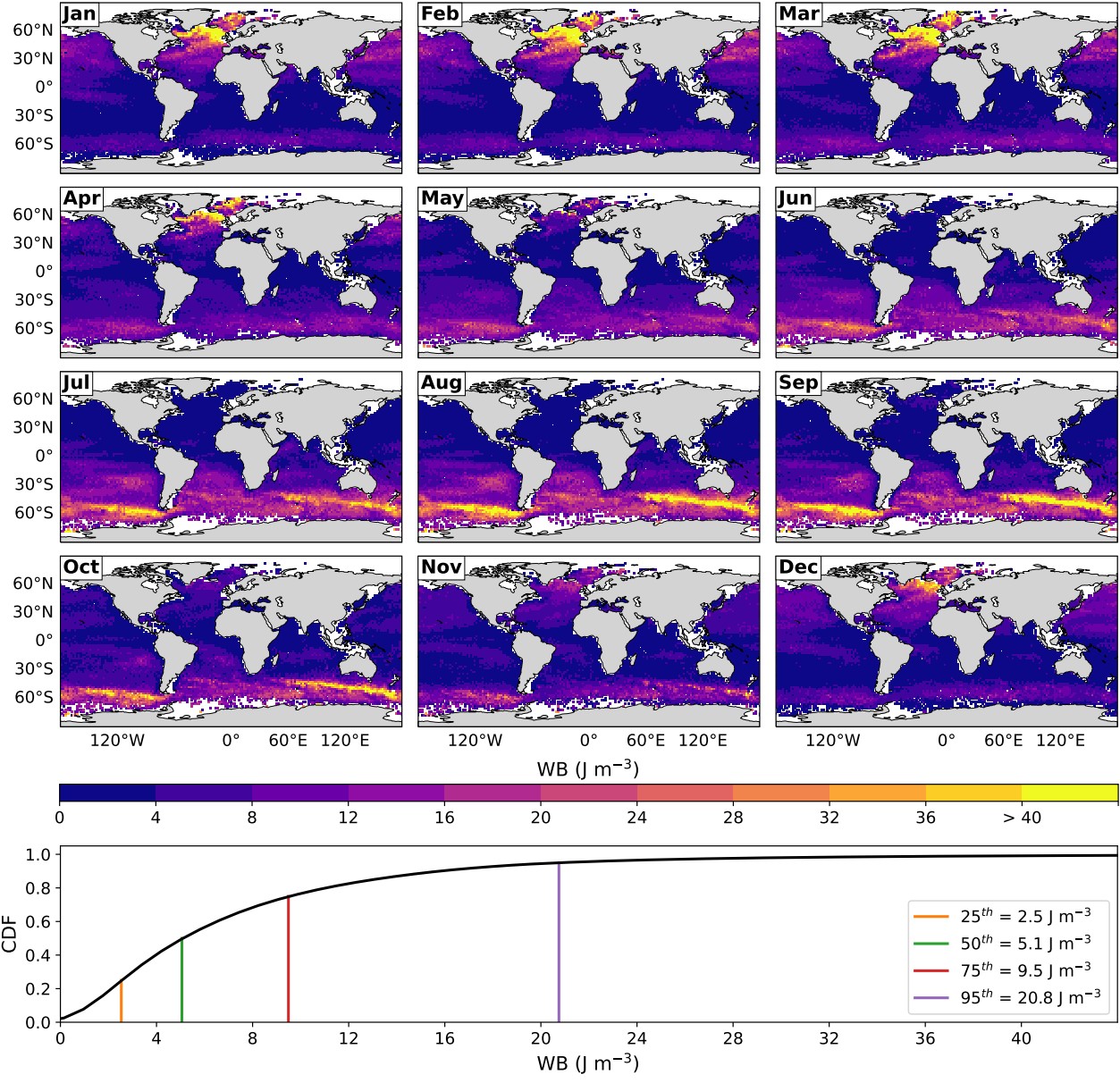

**Figure 4.** Upper panel: global monthly climatology of the WB threshold characterizing the EBM-MLD, i.e., $\mathrm{WB}(z = \mathrm{MLD})$, considering $\Delta\overline{\rho_0^\theta} = 0.0150\,\mathrm{kg\,m^{-3}}$. Lower panel: the cumulative density function (CDF) of the conjoint distribution in space and time of $\mathrm{WB}(z = \mathrm{MLD})$, with various percentiles shown. The maximum WB threshold is $156\,\mathrm{J\,m^{-3}}$.

To explore if the mixed layer is energetically homogeneous, we computed a quality index for WB ($QI_{WB}$) following Lorbacher et al. (2006), who defined it assuming a near-surface layer with quasi-homogeneous properties in which the standard deviation of the property along its vertical mean is close to zero. Following Lorbacher et al. (2006), $QI_{WB}$ can evaluate the degree of homogeneity in WB from the 10 m depth to the MLD and, consequently, the EBM performance in determining the MLD according to the following criteria: $QI_{WB} > 0.8$ indicates a well-homogeneous layer in WB, $0.5 \leq QI_{WB} \leq 0.8$ indicates increased uncertainty in the existence of a quasi-homogeneous layer in WB, and $QI_{WB} < 0.5$ indicates that there is no a quasi-homogeneous layer in WB (a common result for profiles where WB changes gradually with depth). Figure 5 shows the global monthly climatology of $QI_{WB}$ for the EBM-MLD, considering $\Delta\overline{\rho_0^\theta} = 0.0150 \, \mathrm{kg \, m^{-3}}$, along with its corresponding CDF. EBM performs very well in almost all the world ocean year-round: 96.72% of the world ocean has $QI_{WB} > 0.8$, whereas only in 0.03% of the world ocean EBM has $QI_{WB} < 0.5$. During the transition from wintertime to springtime in the North Atlantic and Southern Oceans, $QI_{WB} \approx 0.7$, indicating a reduced but still good EBM performance.

The occurrence of strong convective processes in high-latitude regions during winter generally produces homogeneous layers to very great depths. In these cases, the potential density profile is very smooth, resulting in high values of the MLD and its associated WB threshold. The smoothness of the WB profile, with a structural change hardly identifiable, results in $QI_{WB}$ being close to 1. These profiles pose a challenge to the quality index of Lorbacher when identifying an upper homogeneous layer, making the interpretation of $QI_{WB}$ in terms of its numerical values ambiguous and potentially inaccurate. The difficulty in describing the homogeneity in WB in terms of $QI_{WB}$ in high-latitude regions during winter also applies to the deep-water formation regions. The above results in that the estimation of the WB threshold characterizing the MLD in those regions has a larger uncertainty than that estimated in regions where an energetically homogeneous layer is easily identifiable. Despite the above, we assume that $QI_{WB}$ is enough representative of the vertical homogeneity of the mixed layer in high-latitude and deep-water formation regions; more on this issue is addressed in the Discussion. The most relevant aim of this analysis is to evaluate the general performance of EBM.

The previous analyses showed that EBM provides MLD estimates consistent with the space-time variability in stratification across the world ocean throughout the year, which agrees with the expected physical behavior. The resulting MLD delimitates a well-mixed layer in energetic terms under different ocean conditions in highly and slightly stratified regions, suggesting it can represent a good standard in determining the MLD with global applicability during all seasons. However, it remains to investigate to what extent the layer quasi-homogeneous in energy is homogeneous in density and temperature. Figures 6 and 7 show a monthly climatology of the absolute differences in potential density and conservative temperature from the reference depth of 10 m to the EBM-MLD, respectively; their corresponding CDFs are also shown.

In constructing the mixed layer definition, the density variations along the mixed layer throughout the year were established. The maximum differences in potential density from the reference depth of 10 m to the EBM-MLD are limited by approximately $2\Delta\overline{\rho_0^\theta}$, while the minimum differences are only slightly greater than $\Delta\overline{\rho_0^\theta}$. For the EBM-MLD climatology, considering $\Delta\overline{\rho_0^\theta} = 0.0150 \, \mathrm{kg \, m^{-3}}$, most of the world ocean should have potential density differences within the interval $(0.0150, 0.0300) \, \mathrm{kg \, m^{-3}}$ throughout the year (Fig. 6). Density differences smaller than $0.0150 \, \mathrm{kg \, m^{-3}}$ can occur in regions with very shallow mixed layers where the density profiles are noisy, with values oscillating around the density value at the reference depth. The spatial

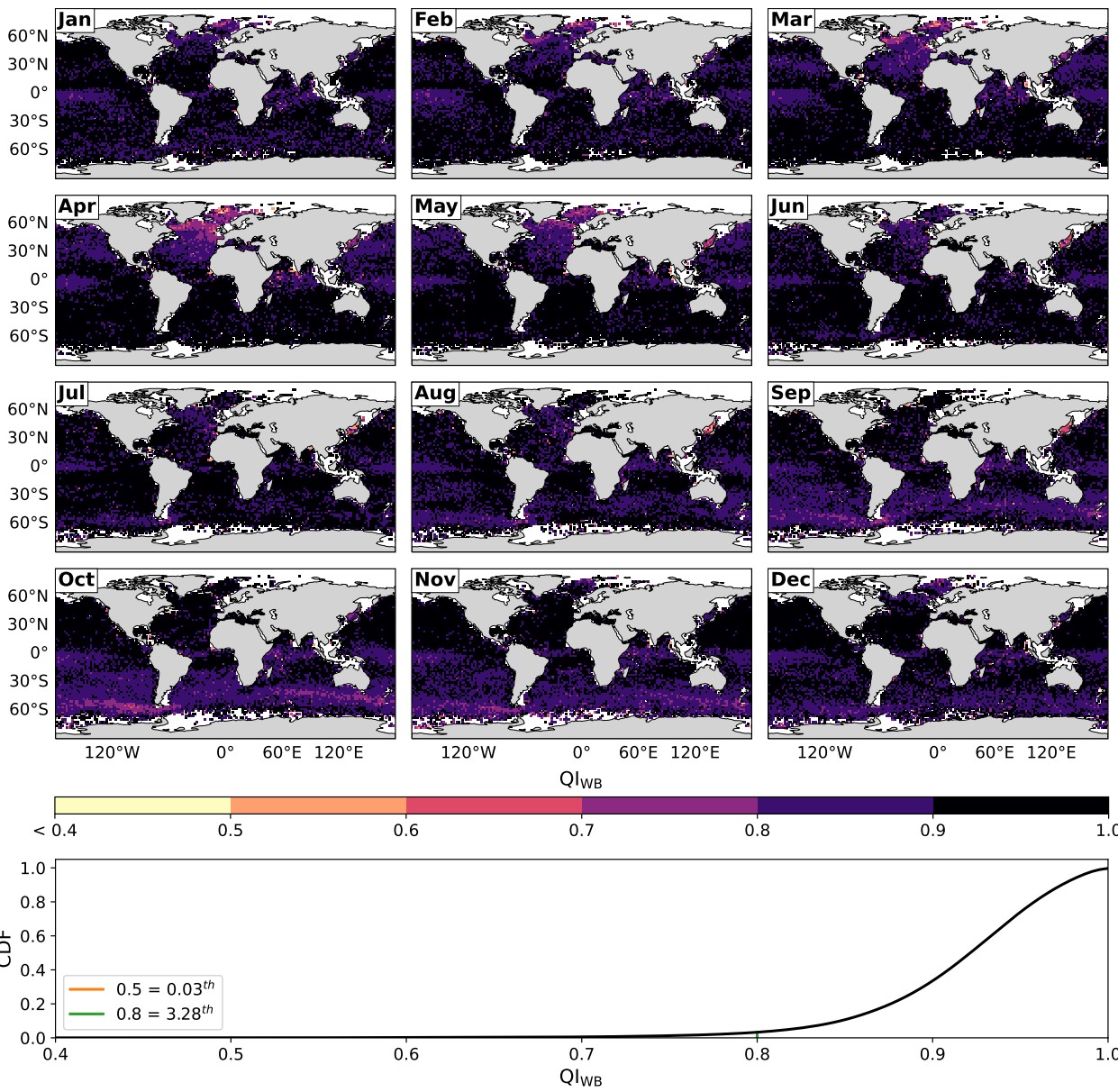

**Figure 5.** Upper panel: global monthly climatology of $QI_{WB}$ for the EBM-MLD, considering $\Delta \overline{\rho_0^{\theta}} = 0.0150$ kg m$^{-3}$. Lower panel: the cumulative density function (CDF) of the conjoint distribution in space and time of $QI_{WB}$, with two $QI_{WB}$ values and their corresponding percentiles shown.

variability of the density differences is not expected to have particular characteristics, such as showing high heterogeneity or seasonal behavior. EBM was constructed to have density variations along the mixed layer that are very restricted globally

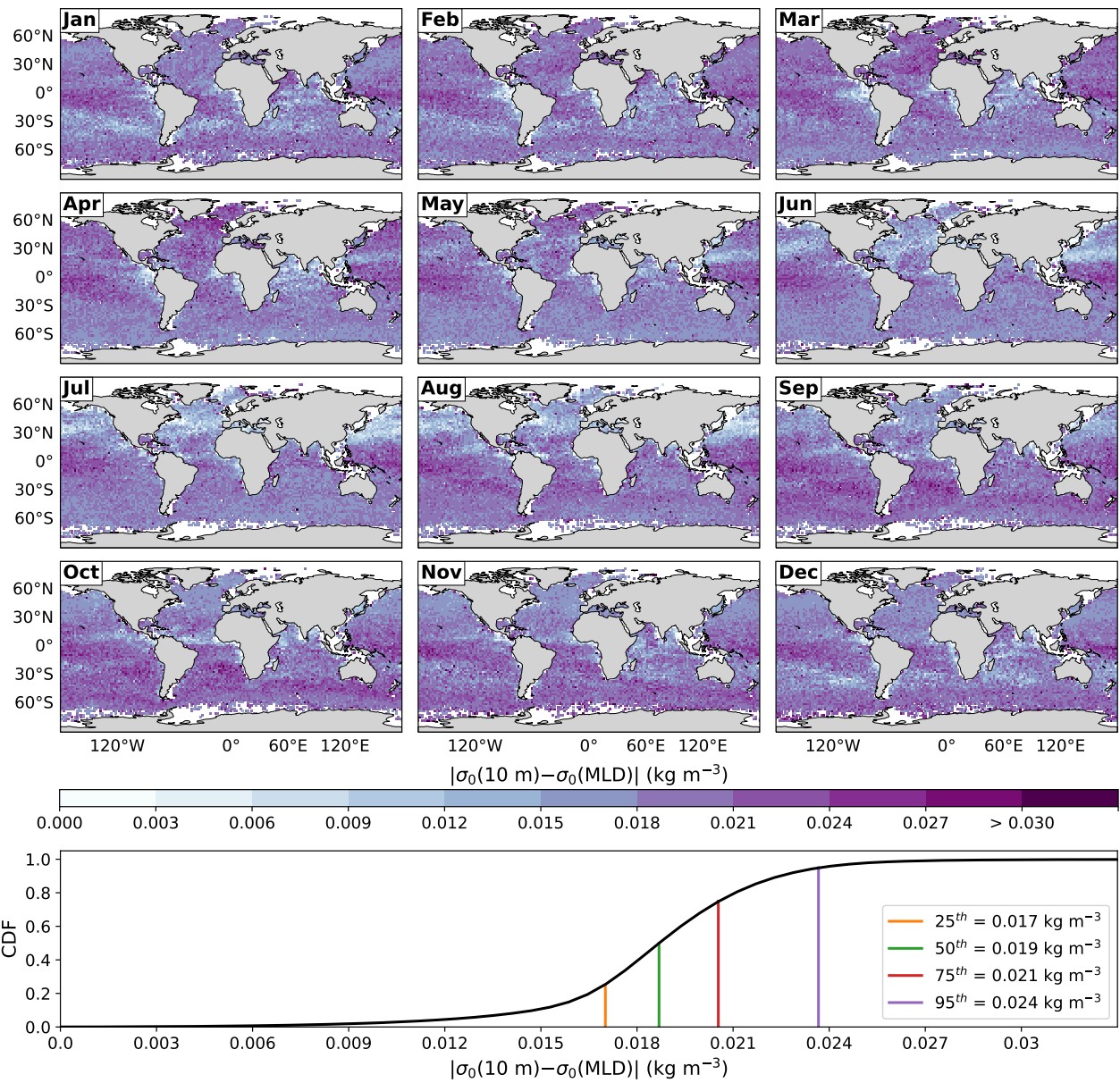

**Figure 6.** Upper panel: global monthly climatology of the absolute differences in potential density from the reference depth of 10 m to the EBM-MLD, i.e., $|\sigma_0(10\,\text{m}) - \sigma_0(\text{MLD})|$. Lower panel: the cumulative density function (CDF) of the conjoint distribution in space and time of $|\sigma_0(10\,\text{m}) - \sigma_0(\text{MLD})|$, with various percentiles shown. The maximum density difference is $0.114\,\text{kg m}^{-3}$.

year-round (see the CDF in Fig. 6). Note that this result does not contradict the expected physical behavior relating the density stratification and the MLD, according to which the stronger the density stratification, the smaller the MLD and vice versa.

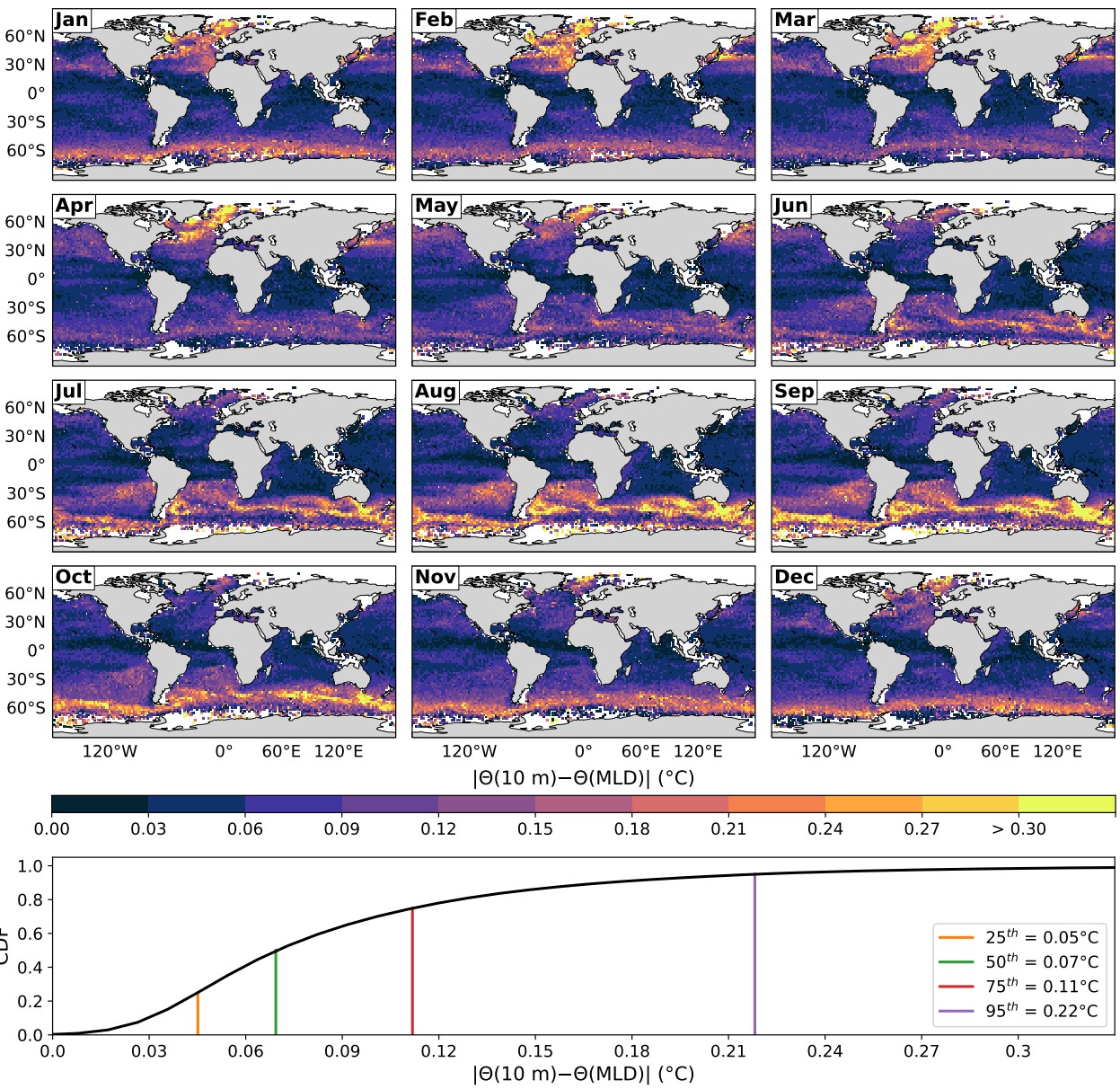

**Figure 7.** Upper panel: global monthly climatology of the absolute differences in conservative temperature from the reference depth of 10 m to the EBM-MLD, i.e., $|\Theta(10\,\text{m}) - \Theta(\text{MLD})|$. Lower panel: the cumulative density function (CDF) of the conjoint distribution in space and time of $|\Theta(10\,\text{m}) - \Theta(\text{MLD})|$, with various percentiles shown. The maximum temperature difference is 1.82°C.

The differences in conservative temperature from the reference depth of 10 m to the MLD shown in Fig. 7 are heterogeneous in space and change over time with a type of seasonal variation. The temperature differences are generally large for large MLDs and vice versa; however, the temperature differences do not have the same structure or seasonal variation as those

of the MLD. The corresponding CDF shows that 95% of the global ocean has temperature differences of less than 0.22°C throughout the year. The most relevant contribution of this figure is the homogeneity in temperature rather than a detailed analysis of the spatiotemporal variability of these differences. In summary, EBM determines a quasi-homogeneous mixed layer in WB, density, and temperature for the global ocean throughout the year, in agreement with our definition and those of de Boyer Montégut et al. (2004).

Finally, we analyzed the global monthly MLD climatology calculated with EBM and its corresponding WB threshold climatology, considering $\Delta\overline{\rho_0^\theta} = 0.0625$ kg m$^{-3}$ (Figs. S3 and S4 in the Supplement). Compared to the case with $\Delta\overline{\rho_0^\theta} = 0.0150$ kg m$^{-3}$, the MLD magnitude is larger, and its seasonal changes are somewhat blurred. The associated WB thresholds are large enough to be representative of layers with small WB values, with 50% of the world ocean having WB thresholds exceeding 30 J m$^{-3}$ year-round. Therefore, it is concluded that the MLD obtained with $\Delta\overline{\rho_0^\theta} = 0.0625$ kg m$^{-3}$ does not produce

a quasi-homogeneous layer in WB and is inconsistent with our mixed layer definition. A corollary of this result is that MLD methodologies based on density thresholds of about 0.125 kg m$^{-3}$ along the mixed layer produce overestimated MLDs and are inadequate to define a well-mixed layer in energetic terms.

## 3.2  MLD methodologies intercomparison

The mixed layer definition depends on the parameter being addressed, which has resulted in numerous MLD methodologies

whose estimates do not completely agree with each other. Figures S5, S7, S9, and S11 in the Supplement show the global monthly MLD climatologies calculated with B04T, B04D, HT09, and R23, respectively. From those climatologies and the corresponding EBM climatology (Fig. 3), we evaluated the conjoint uncertainty in the MLD estimation via a percentage error,

$$\text{MLD uncertainty} = \frac{0.5 \times \Delta\text{MLD}}{\overline{\text{MLD}}} \times 100\%, \tag{6}$$

where $\Delta$MLD is the range in the MLD estimated by the five methodologies, and $\overline{\text{MLD}}$ is the corresponding mean MLD; the

smaller the MLD range, the smaller the MLD uncertainty, and vice versa.

The global monthly climatology of the MLD uncertainty is shown in Figure 8; it exceeds 19% for half of the world ocean throughout the year. All the methodologies approximately coincide in the MLD calculation for profiles near the ideal, that is, for hydrographic profiles with a clear homogeneous upper section and sharp density and temperature gradients below it, without temperature inversions. For profiles with increasing density and decreasing temperature from the near-surface, all the

methodologies yield very shallow MLD values, which do not necessarily coincide. The most significant disparity in the MLD values is obtained for profiles with a quasi-homogeneous upper section and smooth gradients below it. In some cases, some common methodologies fail to provide an MLD value. Large MLD uncertainties are common during winter and spring when mixing is more active, eroding sharp density and temperature gradients in winter and creating near-surface restratification in spring. Large uncertainties are ubiquitous across the ocean; however, they are mainly located in regions where salinity

significantly influences density, such as polar seas, intermediate and deep-water formation regions, and barrier and compensated layers; they are also predominant along the equator.

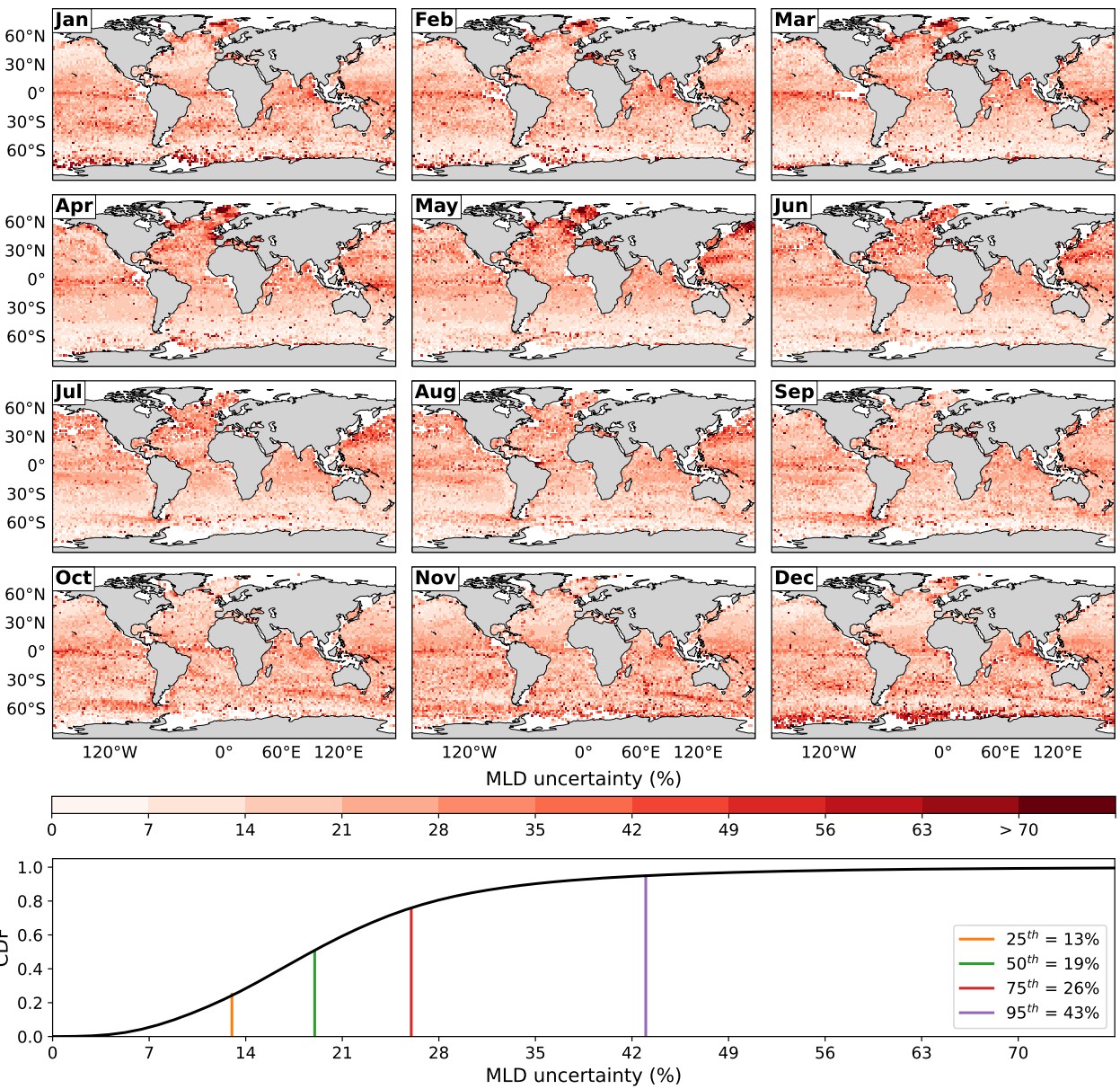

**Figure 8.** Upper panel: global monthly climatology of the conjoint uncertainty in the MLD estimation, considering the methodologies B04T, B04D, HT09, R23, and EBM. Lower panel: the cumulative density function (CDF) of the conjoint distribution in space and time of the MLD uncertainty, with various percentiles shown.

The above analysis underscores the subjective nature of the mixed layer definition and the resulting lack of consistency among the methodologies in determining the MLD. It is almost impossible to ensure that the true value of the MLD in any location and time is known; the different MLD methodologies distinctly evaluate the mixing conditions across the ocean.

Consequently, the accuracy of any methodology can not be determined, that is, the closeness of any MLD estimation to the true value. When comparing MLD methodologies, we can only evaluate their precision: the closeness between the different MLD estimates. The reliability of EBM in determining the MLD was evaluated, to a certain extent, using the quality index $QI_{WB}$. To complete the EBM's reliability evaluation, we evaluated the EBM's precision by comparing its MLD estimates with those of the common methodologies (i.e., B04T, B04D, HT09, and R23). Qualitatively, the structure of the spatiotemporal

variability of the MLD is consistent among all the methodologies (see Figs. 3, S5, S7, S9, and S11). In this regard, EBM is precise compared to the others and provides a realistic description of the MLD, consistent with the seasonal variation of ocean conditions across the ocean.

To quantitatively analyze the EBM's precision, we considered the global PDF and CDF of the MLD for each season, obtained with each methodology (Fig. 9). A clear seasonal behavior of the MLD, with the largest MLDs during wintertime and

the smallest ones during summertime, is consistent across all methodologies. However, the MLD magnitude differs among methodologies, with the smallest MLDs obtained with EBM and the largest ones obtained with B04T (Fig. 9). The global PDF of the MLD for each season was analyzed using some precision measures: median, variance, and skewness. Table 1 shows the value of each precision measure for each season and methodology. To evaluate EBM, we calculated the average value and range for each precision measure considering the four common methodologies; also, we calculated the corresponding

percentage difference of EBM according to the following formula,

$$\text{percentage difference of EBM} = \frac{\text{EBM measure - average value of common methodologies}}{\text{average value of common methodologies}} \times 100\%. \tag{7}$$

The global PDF of the MLD for each season varies among methodologies (Table 1). However, the most relevant contribution of this table is the evaluation of the EBM's precision, measured via its percentage difference. The common methodologies provide an ensemble of the possible extent of the precision measures, in which the smallest and largest values account for the

uncertainty in calculating it (the range). In this ensemble approach, EBM can be considered precise if its values are inside that interval. According to the percentage difference, EBM underestimates the median of the MLD throughout the year, from 11% during winter and fall to 27% during spring; EBM is not precise in calculating the MLD median. Regarding the MLD variance, EBM is precise in calculating it. Finally, EBM underestimates the MLD skewness during summer and fall but provides precise values during winter and spring.

The MLD methodologies are qualitatively consistent in the spatiotemporal variability of the MLD. Quantitatively, EBM is precise in a statistical ensemble sense; regarding the statistical distribution of the MLD, the methodologies agree in their variance but differ in their median and skewness. The above raises the question of whether it is possible to determine the best MLD methodology. All the methodologies perform well under the oceanographic conditions for which they were designed, according to the parameter being addressed; Tang et al. (2025) evaluated 12 MLD methodologies and found that each has

unique merits and limitations that depend on the analyzed ocean conditions. The determination of the best MLD methodology thus depends on the criterion used to rank the methodologies. However, we can take a step further in addressing this question by evaluating the energy-consistency of the MLD methodologies. If the behavior of an MLD methodology deviates from the energy definition of the mixed layer, it is not energy-consistent. According to the aforementioned definition, the mixed layer is

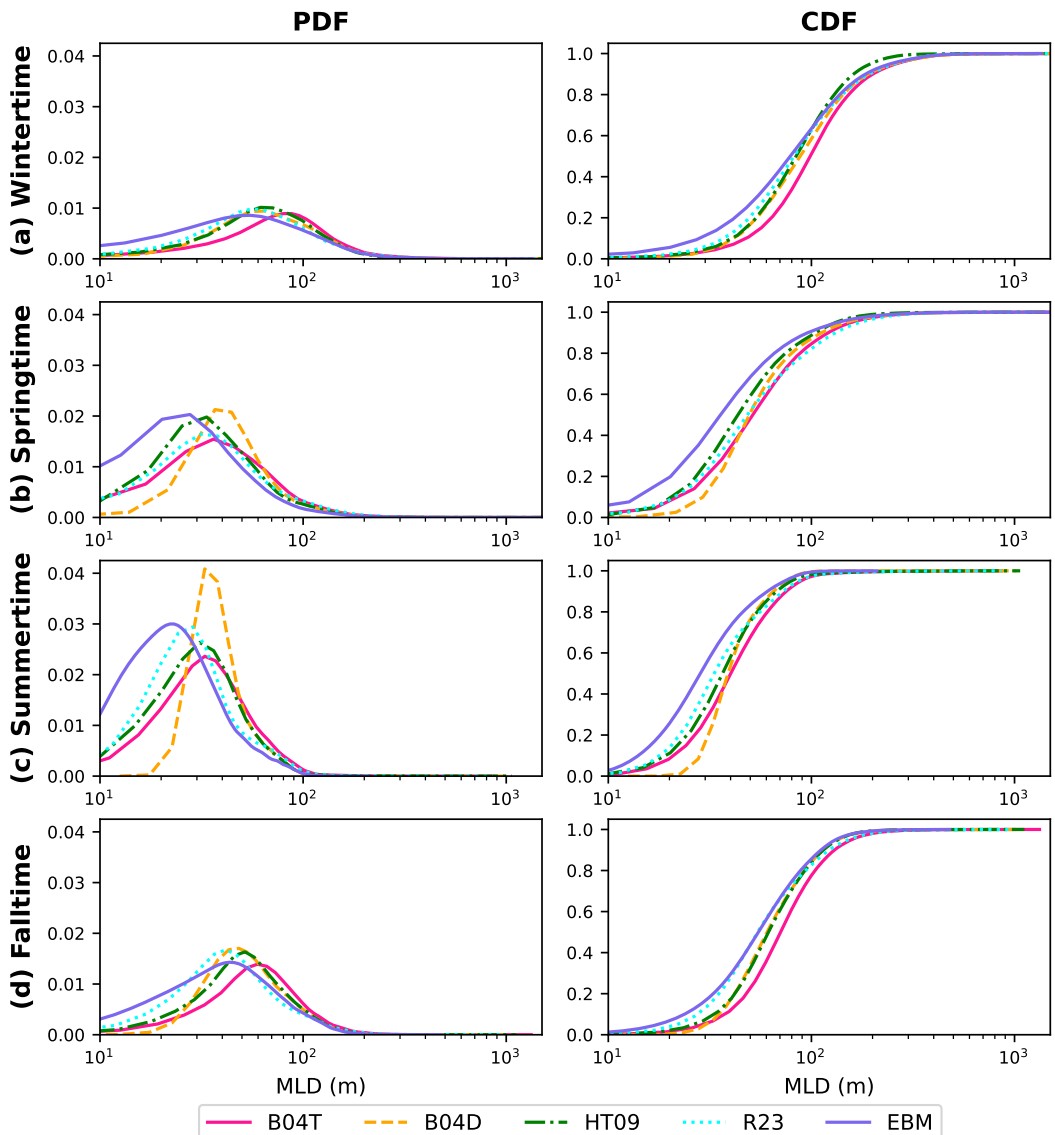

**Figure 9.** The global probability density function (PDF) and cumulative density function (CDF) of the MLD for (a) wintertime, (b) springtime, (c) summertime, and (d) falltime, obtained with each methodology. The PDF and CDF represent the conjoint distribution in space and time of the MLD.

the energetically homogeneous layer characterized by zero or small WB values; moreover, according to Eq. (5), it is acceptable
to have shallow mixed layers associated with small WB values and deep mixed layers associated with large WB values.

To evaluate the energy-consistency of the MLD methodologies, we considered the global monthly MLD climatology obtained with each methodology and calculated the WB value at the MLD (Figs. 4 and S6, S8, S10, and S12 in the Supplement);

**Table 1.** Precision measures (median, variance, and skewness) of the global PDF of the MLD for each season and methodology. The average value and range for each precision measure, calculated considering the four common methodologies (B04T, B04D, HT09, and R23), are also shown. The percentage difference of EBM relative to the different average values of common methodologies is also shown.

| Season | Statistic | B04T | B04D | HT09 | R23 | Average values | EBM (percentage difference) |
|---|---|---|---|---|---|---|---|
| Wintertime | Median (m) | 98 | 88 | 84 | 82 | Average = 88 <br> Range = 16 | 78 (-11%) |
| | Variance (m$^2$) | 5932 | 6807 | 3059 | 5695 | Average = 5373 <br> Range = 3748 | 5888 (+10%) |
| | Skewness | 3.13 | 4.34 | 3.69 | 2.55 | Average = 3.43 <br> Range = 1.79 | 2.57 (-25%) |
| Springtime | Median (m) | 51 | 48 | 43 | 47 | Average = 47 <br> Range = 8 | 35 (-27%) |
| | Variance (m$^2$) | 3204 | 2981 | 1899 | 3403 | Average = 2872 <br> Range = 1504 | 2384 (-17%) |
| | Skewness | 6.7 | 7.86 | 6.25 | 3.77 | Average = 6.14 <br> Range = 4.1 | 4.93 (-20%) |
| Summertime | Median (m) | 40 | 39 | 36 | 33 | Average = 37 <br> Range = 7 | 28 (-23%) |
| | Variance (m$^2$) | 895 | 302 | 810 | 902 | Average = 727 <br> Range = 600 | 360 (-50%) |
| | Skewness | 6.51 | 11.96 | 8.83 | 6.10 | Average = 8.35 <br> Range = 5.86 | 1.44 (-83%) |
| Falltime | Median (m) | 70 | 60 | 61 | 54 | Average = 61 <br> Range = 17 | 55 (-11%) |
| | Variance (m$^2$) | 1966 | 1356 | 1389 | 2135 | Average = 1711 <br> Range = 779 | 1400 (-18%) |
| | Skewness | 3.53 | 3.73 | 3.73 | 3.09 | Average = 3.52 <br> Range = 0.65 | 1.76 (-50%) |

in this way, we were able to analyze their energy-consistency on a global scale throughout the year. The methodologies B04T, HT09, and R23 are not energy-consistent because the spatiotemporal variability of the WB at the MLD is not consistent with that of the MLD throughout space and time. According to our mixed-layer definition, they are not expected to calculate


the MLD accurately based on energy considerations. Figure 10 exemplifies the WB value at the MLD on global meridional transects in the Pacific and Atlantic Oceans during winter and summertime.

B04D has a behavior that is very close to being energy-consistent (Fig. 10). It has the highest concordance with EBM, as both are built from the density; however, B04D has WB values larger than those of EBM. During winter, B04D tends to overestimate the WB threshold (and consequently, the MLD) in latitudes higher than 40°S and 40°N; during summer, it largely coincides with EBM. Of all the methodologies, B04T and HT09 are the ones that are furthest from being energy-consistent. The largest discrepancies between B04T and HT09 with EBM occur during winter throughout almost all latitudes, especially in low and high latitudes; during summer, the discrepancies are concentrated in low latitudes (between 20°S-20°N); in latitudes south of 20°S and north of 20°N, the methodologies are close to each other. R23 has a behavior close to being energy-consistent, although it persistently exhibits large WB values in low latitudes near the equatorial zone every month. During winter, R23 is close to EBM between 40°S-40°N (where the methodology has its best fit); beyond those latitudes, the WB values are larger than those of EBM. During summer, R23 is very close to EBM throughout almost all latitudes, except near the equatorial zone, where its WB values are larger than EBM's.

The above analysis showed that B04D and EBM are energy-consistent, although WB in B04D is almost twice that of EBM in some regions and months, making it difficult to reconcile the large WB values of B04D with our mixed layer definition. By being physically derived and based on energy, EBM could be superior to B04D in estimating the MLD; EBM could represent an improved or well-founded version of the threshold density criterion proposed by de Boyer Montégut et al. (2004) to define the MLD because EBM considers the density vertically integrated. Nonetheless, despite its qualities, EBM has some downsides; the EBM-MLD intrinsically depends on the $\Delta \overline{\rho_0^\theta}$ threshold, which may negatively influence its performance in analyzing highly stratified or vertically compensated layers. It is important to note that the common methodologies can also struggle or even fail when analyzing profiles that strongly differ from the ideal ones (an upper homogeneous layer above a strong pycnocline or thermocline). For highly stratified layers, different $\Delta \overline{\rho_0^\theta}$ thresholds could lead to very different MLDs; however, the requirement of having small WB values could lessen this limitation and restrict the variation in the MLD values. For vertically compensated layers, like B04D, EBM may also overestimate the MLD; nonetheless, using WB, we can measure the degree of inhomogeneity of the water column associated with the compensated layer and investigate whether it is intense enough to suppress mixing. While EBM may not provide a better or more meaningful MLD estimate than other methodologies, it does measure the water column inhomogeneity in terms of energy, a unique feature that other methodologies lack. Furthermore, EBM provides realistic MLD estimates and performs without failure in complex profiles, demonstrating its robustness under different ocean conditions.

Similar to other MLD methodologies, EBM is sensitive to the choice of the reference depth, mainly in regions with very thin mixed layers and during winter and early spring when mixing is more active, eroding sharp density and temperature gradients in winter and creating near-surface restratification in spring. The above is not a limitation for EBM, as it is based on the analysis of WB with depth; EBM can still be used to find the MLD and its associated WB. For those regions and during those periods, the reference depth can be adapted to be consistent with the local dynamics; then, the procedure described in the Methods (section 2) can be applied. We also explored the influence of the vertical resolution of the density profiles on the MLD calculation. We found that as long as the vertical characteristics of the density are correctly resolved and sampled, the

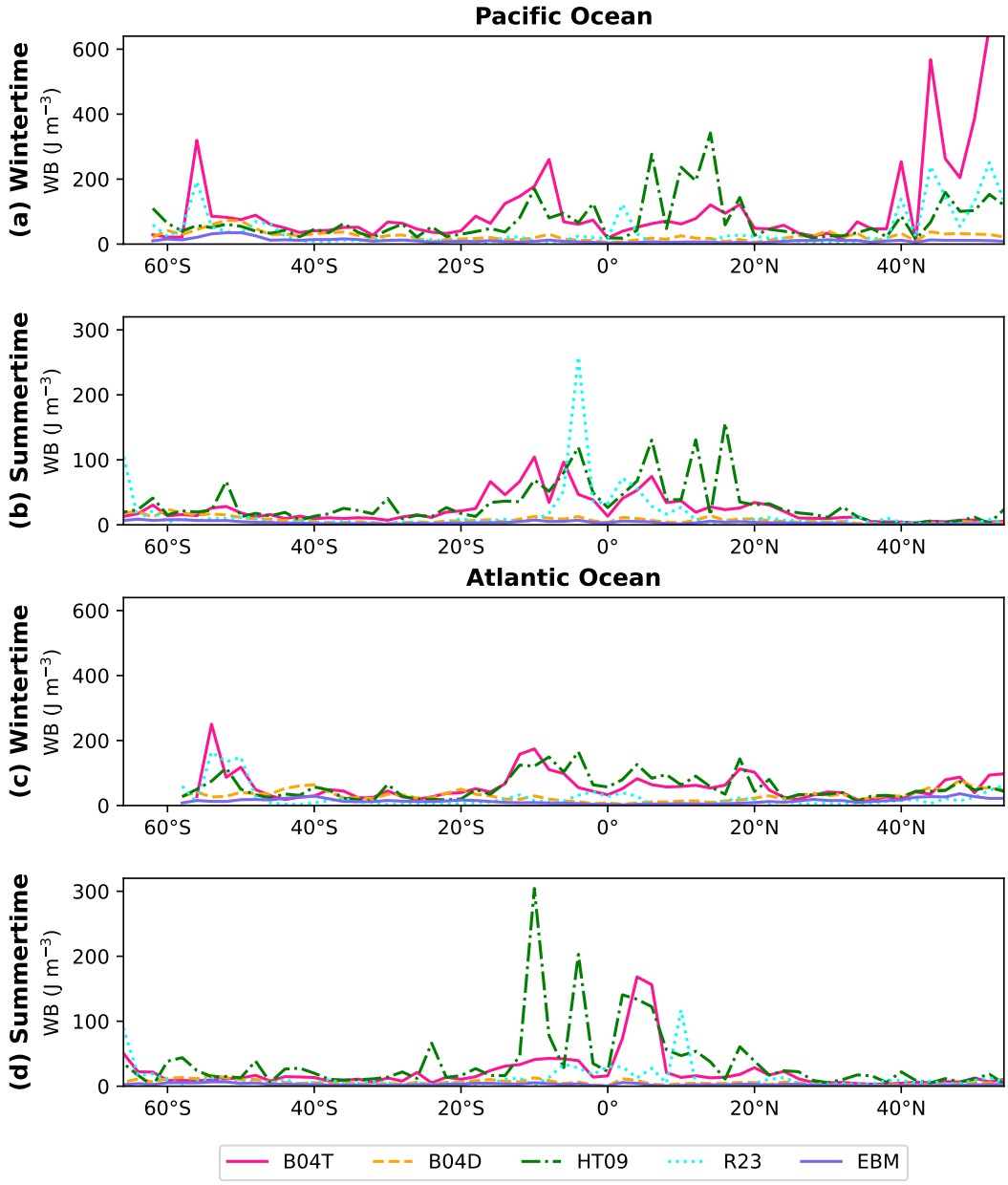

**Figure 10.** The WB value at the MLD for each methodology on global meridional transects in the Pacific (along 150°W) and Atlantic (along 30°W) Oceans during wintertime (February in the Northern Hemisphere and August in the Southern Hemisphere) and summertime (August in the Northern Hemisphere and February in the Southern Hemisphere). The global monthly MLD climatology obtained with each methodology was used for the calculation.

estimated MLD will be accurate to the order of the vertical resolution. This adaptability provides flexibility in applying the methodology in specific regions and under different ocean conditions.

### 3.3 Can a unique WB value define the MLD globally throughout the year?

Our results showed that the mixed layer obtained with EBM is associated with WB thresholds with little variation on a global scale. To a certain degree, the WB threshold is region—and season—independent (Fig. 4), which raises the possibility that a few or even a unique WB threshold can characterize the MLD globally year-round. To what extent is the WB threshold region—and season—independent? To explore this question, we present a very preliminary result delving into whether a unique WB value can define the MLD globally during all seasons, a question posed by Treguier et al. (2023). The above question was addressed by analyzing three global meridional transects in the Pacific, Atlantic, and Indian Oceans during August and February (not shown). For each transect, the EBM-MLD was calculated along with the differences in depth between the MLD and three WB values (Fig. 11). It was found that a unique WB value does not consistently locate the MLD. For latitudes north of 20°S, the 5 J m$^{-3}$ WB isoline well locates the MLD; for latitudes south of 20°S, the 12.5 or 20 J m$^{-3}$ WB isolines are more appropriate. As expected, large WB thresholds characterizing the MLD were found in high-latitude regions during wintertime. However, as described above in Fig. 4, there is uncertainty in estimating the WB threshold in those regions. The impact of that uncertainty on identifying a unique WB threshold for the MLD globally year-round remains unclear, as does whether the difference in WB isolines between winter and summer will increase or decrease. We will investigate this in future work.

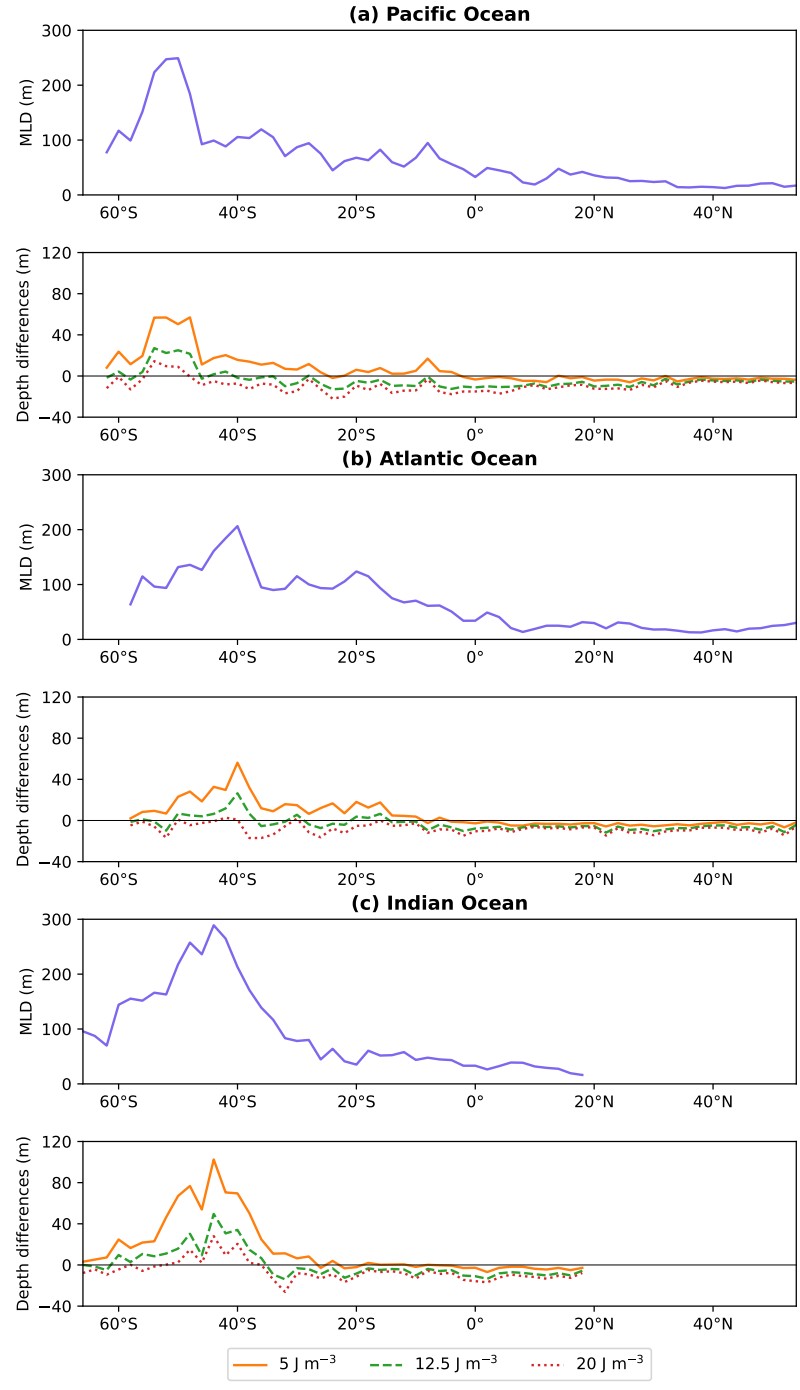

**Figure 11.** The EBM-MLD (solid purple line) along three global meridional transects in the Pacific (150°W), Atlantic (30°W), and Indian (90°E) Oceans during August. The differences in depth between the MLD and three WB equipotentials are also shown for each transect.

The average difference in depth between the MLD and the three WB isolines along each transect is shown in Table 2; for this calculation, we used the absolute values of the differences in depth. Given the small differences in depth of the 12.5 and 20 J m$^{-3}$ WB isolines, some WB equipotential in that interval could be a good choice to delineate an energetically well-mixed upper ocean layer, thereby defining the MLD. The above suggests that a unique WB equipotential could define the MLD throughout the three ocean transects, a remarkable finding that would indicate that the mixed layer is close to being energy-consistent across space and time. Whether this result can be extended and applied on a global scale during all seasons is an endeavor beyond the scope of this study that deserves further research.

**Table 2.** The average difference in depth between the EBM-MLD and the three WB isolines along each transect shown in Fig. 11. For this calculation, the absolute values of the differences in depth were used.

| Ocean | 5.0 J m$^{-3}$ | 12.5 J m$^{-3}$ | 20.0 J m$^{-3}$ |
|---|---|---|---|
| Pacific | 9.3 m | 7.0 m | 9.6 m |
| Atlantic | 8.4 m | 6.2 m | 8.2 m |
| Indian | 17.2 m | 10.1 m | 9.5 m |
| Average value | 11.6 m | 7.7 m | 9.1 m |

## 4   Discussion

In this study, we present a methodology for calculating the MLD, based on physical principles and energy considerations, which promotes the development of energy-based methodologies. The developed methodology, EBM, identifies the upper section of the ocean, well-mixed in energetic terms, in which water parcels can move with little or no WB, which can be considered in contact with the atmosphere and thus be referred to as the mixed layer. EBM uses WB and considerations about the density structure of the water column to define the MLD. The most important characteristic of EBM is that it provides realistic MLD estimates in all world regions and performs without failure in complex profiles, demonstrating its robustness under different ocean conditions (subsection 3.1). We showed a plausible connection between WB and the turbulent kinetic energy budget of the upper ocean layer, suggesting that EBM is consistent with the turbulent approach to the mixed layer formation (D'Asaro, 2014; Sutherland et al., 2014; Franks, 2014; Sallée et al., 2021). The mixed layer is thus determined by energy processes instead of density, temperature, or salinity thresholds, which vary in space and time according to the oceanographic conditions of the study region (Griffies et al., 2016; Treguier et al., 2023). EBM can be the base methodology for performing MLD model intercomparison studies, as in the OMIP and CMIP projects (Griffies et al., 2016; Treguier et al., 2023). The numerical implementation of EBM only requires the potential density profile referred to 0 dbar, which is easily obtained from simple survey ocean data or numerical data. The script to compute the MLD is very short, and its formulae are not complex. In that regard, EBM is easy to implement numerically.

The common and energy-based methodologies use a threshold to define the MLD. However, the nature of their thresholds is substantially different: common thresholds only consider the difference in values of a specific oceanic variable between two

depths, whereas energy thresholds consider the cumulative effects of those differences along the vertical. As shown in this study and by Reichl et al. (2022), mixing resistance depends on the differences in density and the physical distance between two depths. EBM is based on a threshold in WB and is more than a common threshold. While the buoyancy force is directly proportional to the difference in density between two depths, the associated work is not. That means that the difference in density between two given depths cannot be used as a proxy for the energy required to homogenize the ocean's upper layer. Similarly, density-derived measures of the local stability or homogeneity of the water column, such as the buoyancy frequency, cannot be used as proxies for the energy required to homogenize the water column if they do not consider the cumulative effect of the buoyancy force along a water column section. Therefore, the density threshold, density gradient, and buoyancy frequency criteria may not adequately address the mixing resistance when calculating the MLD, as has been done in previous research (Lukas and Lindstrom, 1991; Large et al., 1997; de Boyer Montégut et al., 2004; Lorbacher et al., 2006; Dong et al., 2008; Holte and Talley, 2009; Chu and Fan, 2011; Carvalho et al., 2017).

Reichl et al. (2022) introduced a framework for defining the MLD based on the PE anomaly, suggesting a threshold range of 10-25 J m$^{-2}$. While this range offers a valuable benchmark, determining the optimal thresholds under all seasonal and regional conditions remains an open question. They found that a spatially and temporally variable threshold in the PE anomaly should be used to reproduce, to some extent, MLDs similar to those obtained with HT09. However, since HT09 is non-energy-based, attempting to match the performance of an energy-based methodology with that of a non-energy-based one may not be meaningful. Future research could determine optimal values of the PE anomaly for global application and explore this approach in regional contexts by assessing the sensitivity of the MLD estimates to different thresholds of the PE anomaly. By comparison, with EBM, we were able to find the energy values (in terms of WB) that define the MLD globally during all seasons. Moreover, EBM provides additional information not supplied by the existing MLD methodologies; WB represents the potential energy barrier to the vertical displacement of water parcels, which could complement the analyses of oceanic processes occurring in intermediate vertical sections, commonly associated with interchanges of properties along the water column, such as the flux of particulate organic matter from the surface to sediments (Kirillin et al., 2012; Omand et al., 2020), the vertical content of chlorophyll (Carvalho et al., 2017; Briseño Avena et al., 2020), and entrainment in barrier layers (Katsura et al., 2022). The application of WB to analyze those processes is beyond the scope of this study and is proposed for future research.

The definition of the mixed layer as the ocean's surface layer whose properties (density, temperature, salinity, and other tracers) are relatively homogeneous in the vertical is challenging to achieve when considering constant increases in density or constant decreases in temperature from the corresponding values of these variables at the reference depth. Since the coefficients of the equation of state of seawater vary with pressure, temperature, and salinity, a given density change does not correspond to a unique temperature change, and vice versa. To determine a homogeneous mixed layer from density or temperature thresholds, the thresholds should vary according to the oceanographic conditions of the study region; moreover, implementing spatially variable thresholds in a set of models and observations would be complex and daunting (Griffies et al., 2016; Treguier et al., 2023). The above supports the definition of the mixed layer as the ocean's upper layer, quasi-homogeneous in buoyancy energy, even if that leads to spatially variable increments in density and decrements in temperature. According to Levitus (1982) and

Kara et al. (2000), variations of up to 0.125 kg m$^{-3}$ in density and up to 0.8°C in temperature can be considered typical in a well-mixed layer. Although our methodology does not seek to determine mixed layers homogeneous in density or temperature, the energy-based mixed layer is very close to such quasi-homogeneity: almost 100% of the world ocean has density differences of less than 0.03 kg m$^{-3}$, and 95% of the world ocean has temperature differences of less than 0.2°C throughout the year.

EBM has several interesting qualities; however, it has some downsides and room for improvement. This study analyzed the MLD on long spatial and temporal scales: spatial scales larger than mesoscale and timescales longer than diurnal cycles. Active mixing and high-frequency MLD variability, mainly driven by synoptic atmospheric forcing, ocean eddies, and fronts (Brainerd and Gregg, 1995; Whitt et al., 2019), were not addressed. The surface turbulent boundary layer can be a more relevant measure to explore the above processes. In computing monthly MLD values from daily values, the sub-monthly variability was omitted, potentially underestimating the MLD compared to the corresponding daily MLD values, as shown by Toyoda et al. (2017). A thorough analysis of regional differences between the monthly and daily MLD values is out of the scope of this study and is proposed for future research. Additionally, due to limitations in the spatial coverage of Argo data, this study could not explore the MLD in coastal zones, and the robustness of the findings in the subpolar oceans may be limited; for future research, we propose incorporating additional observational datasets covering the regions not extensively mapped by Argo to expand the scope and robustness of this study. Additionally, it would be instructive to extend the intercomparison of MLD methodologies by incorporating additional methodologies, such as those analyzed by Tang et al. (2025), who found that the linear fitting method of Chu and Fan (2010) resulted in the most robust approach for calculating the MLD.

Recent research has highlighted temperature inversions as a significant limitation of various MLD methodologies, not only those based on temperature, which restricts their application in regions where temperature inversions are common (Tang et al., 2025). However, density-based methodologies, such as EBM, could have an advantage over temperature-based ones because they adequately incorporate the effects of temperature on mixing conditions into density via the equation of state of seawater. Consequently, EBM could adequately account for the effects of temperature on mixing and in the MLD calculation. Although the mixed layer has been commonly described in terms of physical variables (such as temperature or density), ecological and chemical variables (like chlorophyll and oxygen) are also highly relevant in evaluating mixing conditions along the vertical (Sutherland et al., 2014; Tang et al., 2025). The performance of EBM in calculating the MLD and the associated vertical distribution of different ecological and chemical variables is proposed for future research.

Because the turbulence and its associated energy levels are spatially and temporally variable on a global scale, the water column's stratification and vertical homogenization are not spatially uniform throughout the seasons. Globally, the mixed layer is not associated with a unique density threshold, as evident in Table 1 of Kara et al. (2000), de Boyer Montégut et al. (2004), and Peralta-Ferriz and Woodgate (2015); the compensated layers exemplify that a unique density threshold is inappropriate for the world ocean (de Boyer Montégut et al., 2004). Therefore, the $\Delta\overline{\rho_0^\theta}$ threshold is not expected to be globally uniform year-round, and WB's associated spatial distribution across time remains an open question. A very preliminary analysis of the energy levels at the mixed layer base suggested that a unique WB equipotential in the interval 12.5-20 J m$^{-3}$ could define the MLD globally year-round. Testing the hypothesis that a few or even a unique WB threshold can characterize the MLD globally year-round could determine if the mixed layer is energetically consistent across space and time. Such a study would

require long-term data and a regionalization of the WB thresholds on a global scale. Exploring this hypothesis is an endeavor that deserves further research, as it could enhance our understanding of the mixed layer and the various ocean-atmosphere phenomena to which the MLD is relevant.

The EBM-MLD depends on the choice of the WB threshold, which we set based on a $\Delta \overline{\rho_0^\theta}$ threshold. A significant improvement for EBM would be constructing a criterion to unequivocally determine the WB threshold characterizing a well-mixed layer independently of density. A mathematical problem of this nature would lead to trying to find the solution of only one equation with two unknowns (Eq. 5), an ill-posed problem. Solving this problem is not trivial because we must introduce an additional condition or equation to ensure uniqueness and make the problem well-posed. In the absence of an additional equa-

tion to determine the unique solution for WB, the choice of the value of the remaining variable would be subjective or, at least, based on experience (like in the common threshold MLD methodologies). However, from the energy definition of the mixed layer, we can explore some geometric methods to determine the layer quasi-homogeneous in energy with small WB values without specifying an associated density variation. Methods like the quality index of Lorbacher et al. (2006) or the maximum angle method of Chu and Fan (2011) could be helpful; however, they assume a structural change in the variable of interest

and are not suitable for strongly stratified or very smooth density profiles in which a structural change is difficult to find. For these profiles, a specific WB threshold characterizing a quasi-homogeneous layer in energy is needed. Additional research is required and proposed for future studies concerning the physical properties of WB and the values that accurately determine the vertical extension of the mixed layer.

## 5    Conclusions

Recent research has proposed energy-based methodologies as the best option to calculate the MLD, as they can provide accurate estimates while maintaining the calculations without the unnecessary complexities of the turbulent mixing theory. We contribute to the development of energy-based methodologies to define the MLD. Based on energy considerations, we proposed an MLD methodology that is globally applicable and produces realistic estimates of the MLD. This MLD methodology performs robustly across different regions and ocean conditions, including polar seas, intermediate and deep-water formation

regions, and barrier and compensated layer regions. The mixed layer, determined by energy processes, is quasi-homogeneous in energy, density, and temperature in most of the global ocean throughout the year. A practical contribution of our study is an observation-based global MLD climatology, computed from approximately 2 million delayed-mode Argo profiles. This climatology, useful for studies spanning seasonal to climate time scales, from regional to large spatial scales, can also serve as a reference for validating Oceanic General Circulation Model solutions and performing MLD model intercomparison studies.

Currently, we are working on investigating the potential of the proposed MLD methodology to better interpret various dynamic (e.g., vertical exchanges within the ocean and between the ocean and the atmosphere), thermodynamic (e.g., upper ocean heat content), and ecological (e.g., chlorophyll-$a$ content and phytoplankton dynamics) processes at regional and global scales.

*Code and data availability.* The methodology presented in this study is licensed under a GNU General Public License v3.0. The source code is available at https://doi.org/10.5281/zenodo.14531829. The latest package version is v1.1.0. The dataset containing the monthly climatology of mixed layer depth and the derived variables calculated in this study is publicly available in SEANOE at https://doi.org/10.17882/106181.

*Author contributions.* EM conceived and designed the study, developed the methodology, wrote the scripts, interpreted the results, wrote and revised the manuscript, and acquired funding. ER designed the study, wrote the scripts, analyzed the data, interpreted the results, created the figures, and wrote and revised the manuscript. KR-M designed the study, developed the methodology, wrote the scripts, interpreted the results, and wrote and revised the manuscript. LT-F designed the study, interpreted the results, and revised the manuscript. All authors contributed substantially to this work and approved it for publication.

*Competing interests.* The authors declare that they have no conflict of interest.

*Acknowledgements.* The authors acknowledge León Felipe Álvarez-Sánchez for his support in preparing the R source code for the methodology on the GitHub platform and in creating the NetCDF file for the SEANOE repository. We thank the editor, Anne-Marie Treguier, and the valuable feedback and suggestions from B. Reichl, H. Giordani, and an anonymous reviewer, which improved this study. UNAM-PAPIIT IN110925 supported this work. The profile data used to compute these climatologies were collected and made freely available by the International Argo Program and the national programs that contribute to it (https://argo.ucsd.edu, https://www.ocean-ops.org). The Argo Program is part of the Global Ocean Observing System.

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
