# Peer review of "The global ocean mixed layer depth derived from an energy approach"

_EGUsphere, 2024_

## Author Comment (AC1)

**Review of the manuscript: https://doi.org/10.5194/egusphere-2024-4079**

**Title: The global ocean mixed layer depth derived from an energy approach**

**Reviewer #1**

We appreciate all your comments. After a careful revision that incorporated your comments and those of the other two reviewers, this version has greatly improved. We trust that you will find the revisions substantial and consider our study a valuable contribution to mixed-layer depth research. Below is a description of the major modifications made to the manuscript:

- The methodology (subsection 2.1) was extended by including details concerning the connection of the work done by buoyancy (WB) with the turbulent kinetic energy budget.
- We revised our definition of the mixed layer and how to calculate its depth (subsection 2.2. Defining the mixed layer). The mixed layer depth (MLD) was not calculated with a unique WB equipotential.
- Since we adjusted the MLD calculation, we rewrote the Results and Discussion sections entirely. The Conclusions and Abstract were also rewritten accordingly.
- The Supplement was rewritten to provide a detailed comparison of our MLD methodology with the common ones; we included global monthly climatologies of the MLD and the WB at the MLD.
- The dataset containing the monthly climatology of mixed layer depth and the derived variables calculated in this study is now publicly available at SEANOE (https://doi.org/10.17882/106181).

Responses to specific comments are shown below.

**Summary**

**Comment 1:** The MLD is not an objective quantity, it requires defining timescales and spatial scales since the ocean surface boundary layer is under consistent and variable external forcing. Furthermore, identifying a mixed layer invokes some qualitative analysis to quantify what is meant by "mixed". This ambiguity makes the present goal of deriving a "global ocean" mixed layer depth challenging, and I acknowledge the efforts the authors have devoted to this matter. I find the proposed WB method for estimating MLD is potentially interesting, but the paper presently relies heavily on ad-hoc and empirical arguments. This limitation makes it difficult to judge the value of the WB approach versus other approaches. The study would therefore benefit significantly if it can identify quantifiable metrics to bolster its claims.

**Answer 1:** We agree with you on the challenges in defining the MLD. In this version, we clarify and better describe the aspects of the mixed layer addressed in our study in *subsection 2.2 Defining the mixed layer (page 5, lines 124-129)*.

We really appreciate that you find our approach to estimating the MLD interesting. We acknowledge that in the prior version of the manuscript, we derived our results based on empirical and non-robust arguments. In *subsection 2.2*, we proposed a quantitative procedure to calculate the MLD to fix that (*page 5, lines 129-133 and pages 7-8, lines 156-184)*.

Due to the adjusted definition of the mixed layer, the new results are completely different from those of the prior version of the manuscript. We conducted a detailed analysis (qualitative and quantitative) of the resulting MLD climatology and the WB threshold that defines it. We also compared our methodology with others on a global scale, considering different metrics. We consider that we greatly improved the significance and interpretation of our results (*Results and Discussion sections*).

**Comment 2:** The desire for a MLD metric led to proposing PE anomaly in the Reichl et al. (2022) analysis of MLDs, as it quantifies the energetic distance of a column of sea water from being well mixed. This naturally leads to the PE anomaly as a possible basis for identifying the MLD, but it is not obvious that the same energetic distance should define the MLD in all regions or on all timescales. WB itself may provide another interesting metric for quantifying mixed layers, but it differs in important ways from the PE anomaly. WB provides an integrated measure of stratification of the column by considering the displacement of a particle from the mixed layer base. This yields some insight into the stratification, but less information than if you evaluated the buoyancy work associated with displacement of all particles within the mixed layer. It is also not obvious that a single WB quantity should define the MLD at all locations and on all timescales.

**Answer 2:** You are right; WB differs from the PE anomaly in that they describe different physical variables. However, we consider that WB is physically suited to calculate the MLD (*subsections 2.1 and 2.2*). Via the WB profile WB($z$), we actually provide the work done by buoyancy to vertically displace any water parcel within the mixed layer to the top of the mixed layer. We used the WB value at the MLD as the energy threshold characterizing the mixed layer.

We agree that it is not obvious that a unique WB equipotential should determine the MLD globally year-round, which was obtained from Eq. 8 and represented in Fig. 2; please refer to the detailed analysis of the WB metric in *subsections 2.1 and 2.2*. Results of the WB threshold characterizing the MLD are shown in *Figs. 4 and 11*.

**Major concerns**

**Comment 3:** The WB method for estimating MLD has merit over a density threshold method since it invokes a dynamical quantity. However, I am unsure that this dynamical quantity truly addresses two main shortcomings of the threshold method. It would help if this paper better identified what it does and doesn't improve upon other methods.
- WB does not offer any physical guidance for choosing a threshold value, so the threshold remains ad-hoc. The paper argues that 20 J/m3 is a universally applicable value, but this was derived empirically based on arguments about its vertical gradient over limited regions of the ocean. There is no physical significance offered for the integrated buoyancy work of 20 J/m3, which would justify it from first principles for identifying the upper ocean mixed layer in all seasons and locations.
- The new method is still sensitive to the details of the choice of the threshold value and the reference depth. Interestingly though, some of this sensitivity is reduced since WB is an integrated quantity and therefore less responsive to noise in a profile.

**Answer 3:** Considering this and the previous comment, we conducted a detailed analysis of the WB metric and its physical significance (*subsections 2.1 and 2.2*). We also revised our definition of the mixed layer and proposed a quantitative procedure for calculating its depth, including details concerning the density inhomogeneity along the mixed layer, the WB threshold, and the reference

depth (*subsection 2.2*). The *Discussion section* extends the analysis of our MLD methodology and its downsides.

**Comment 4:** I did not agree with the claim in the text: "There is a correspondence between our methodology and that of Reichl et al. (2022), suggesting that our energy-based methodology is consistent with the turbulence approach of the mixed layer formation". I do not find the correspondence between the WB and PE anomaly to be obvious, other than both using energy based criteria. PE anomaly is a column integrated energy value that quantifies the distance of the column from being perfectly homogenous, hence it has units of J/m2. WB only considers the energetics of a single parcel advected through the column, hence it has units of J/m3. One could construct different idealized profiles that give the same WB MLD yet have different PE anomalies, which was identified as a shortcoming of other MLD methods in Reichl et al. 2022. Perhaps the WB method has less PE anomaly sensitivity than other methods, but an evaluation of WB in terms of PE anomaly was not attempted in this paper. These important differences from the PE anomaly do not support the statement that this method connects to boundary layer turbulence in the same way as argued for PE anomaly.

**Answer 4:** We are very sorry for this inaccuracy. Now, in *subsection 2.1 An energy measure of the vertical homogeneity of the water column*, we show an alternative expression of WB that allows us to appreciate its connection with the turbulent kinetic energy budget and, thus, with the physics of boundary layer turbulence. We did not provide further analysis of such a connection because that is beyond the scope of this paper and is proposed for future research (*pages 4-5, lines 93-113)*.

Via the WB profile WB($z$), we actually provide the work done by buoyancy to vertically displace any water parcel within the mixed layer to the top of the mixed layer. We used the WB value at the MLD as the energy threshold characterizing the mixed layer.

**Comment 5:** The article claims the WB MLD with an energy value of 20 J/m3 is "accurate, robust, and of global applicability". This claim is based on looking at "challenging regions", which are defined where a suite of existing methods disagree with each other in MLD magnitude. One issue is that this approach appears to weight the analysis to deep MLD regions (usually convective regions), but this appears to deemphasize shallow MLD regions (e.g., the Arctic Ocean (more on this below) and summer time mixed layers in biologically productive regions) that are just as valuable. Furthermore, I did not find a convincing and quantitative argument in section 3.2.2 for why the WB method estimates a better and/or more consistent MLD than other methods in these regions; it mostly relies on ad-hoc arguments.

**Answer 5:** We agree with you that local results can not support global conclusions. To circumvent this, we compared (qualitatively and quantitatively) our methodology with others on a global scale, considering different metrics (*subsection 3.2. MLD methodologies intercomparison*). We removed the limitation of focusing only on "challenging regions". We found that there is a lack of consistency among the methodologies in determining the MLD, but also that all the methodologies perform well under the oceanographic conditions for which they were built, according to the parameter being addressed. Since it is almost impossible to ensure that the true value of the MLD in any location and time is known, the accuracy of any methodology can not be determined, that is, the closeness of any MLD estimation to the true value. When comparing MLD methodologies, we can only evaluate their precision: the closeness between the different MLD estimates. Thus, we evaluated the precision of our methodology on a global scale and removed ad-hoc arguments from the description of our results.

**Comment 6:** I found the analysis of the climatology lacked a significant new/novel result that advances beyond previous work on mixed layer climatologies. It would be useful if issues in previous climatologies could be identified and to discuss why the WB method was able to provide a more meaningful estimate of a MLD climatology (or for what contexts it may be more meaningful).

**Answer 6:** Our study's significance relies on the energy definition of the mixed layer, which represents an important advancement in this topic (*subsections 2.1 and 2.2*). In the revised version of the manuscript, we carried out an intercomparison of MLD methodologies (*subsection 3.2. MLD methodologies intercomparison*) that showed to what extent our MLD estimation could be more meaningful than others; such an analysis was further extended in the *Discussion section*.

**Comment 7:** The article states that it offers globally applicable mixed layer depths and provides a mixed layer depth climatology. Constructing a gridded mixed layer depth climatology requires many decisions, and little technical detail on how this is done was provided. Nor is the resulting climatology data made available, which would severely limit any potential impacts of this work. The climatology is produced only using Argo data, which neglects a lot of additional oceanic profile data that exists. This also leads to coastal oceans and the entire Arctic Ocean being absent from this work.

**Answer 7:** In the revised version of the manuscript, we rewrote how we constructed the MLD climatology, the data we used, and its limitations in spatial coverage (*subsections 2.3 and 2.4, pages 8-9, lines 185-214*), hoping to be clearer. We acknowledge the limitations in the spatial coverage of Argo data; this study could not explore the MLD in coastal zones, and the robustness of the findings in the subpolar oceans may be limited. For future research, we propose incorporating additional observational datasets covering the regions not extensively mapped by Argo to expand the scope and robustness of this study.

The dataset containing the monthly climatology of mixed layer depth and the derived variables calculated in this study is now publicly available at SEANOE (https://doi.org/10.17882/106181). This information was added to the *Data availability section*.

**Other Comments**

**Comment 8:** What reference pressure is used to define the potential density in equation 3? Can a fixed reference pressure be used as a suitable approximation?

**Answer 8:** We are sorry for not providing that information. According to Stewart (2008), the potential density referred to as 0 bar is adequate for vertical displacements not exceeding a few thousand meters (*Page 3, lines 77-79*).

**Comment 9:** The WB method may offer some physical insights into when and why different MLD methods differ, and it is closely related to the threshold method such that it may better ground the threshold values used in present studies. One can rewrite the WB criteria (equation 3, by rewriting the 2nd terms RHS w/ H defined as the MLD and _Hmean defined as the depth integral mean of potential density over the MLD): WB = g*H*[_Hmean - rho_ref], which is now expressed similarly to the threshold method: delta = [rho(z) - rho_ref]. This makes two notable differences of WB from the threshold method more apparent: (1) the density difference is now from the average of the density over the column instead of at the MLD, and (2) the density difference is effectively weighted by the

MLD (and gravity). These have some interesting implications: (1) integrating the potential density makes the MLD less sensitive to noise in the profile, and (2) weighting the threshold by the MLD reduces the depth of deeper MLDs with the same energy value. The first effect seems advantageous for practical use. The second effect makes sense from the perspective that it takes more energy to raise a parcel from a deeper depth, but since deeper MLDs often experience more turbulence and mixing, it is not straightforward to me that the second effect is obviously "better" for MLD identification. The ocean does not experience uniform levels of energy inputs to turbulence. (This is a main reason I am not convinced there should be a single universal energy value for either the WB method or the PE anomaly method.) Furthermore, the 2nd point above about MLD weighting also offers an interpretation of why shallower depths are associated with larger density differences, e.g. Figure 7.

**Answer 9:** We agree with you that our methodology is a threshold method. The procedure to determine the MLD in terms of WB is based on a threshold of the density inhomogeneity along the mixed layer (*subsection 2.2. Defining the mixed layer*). By considering the vertical integration of density, our methodology could represent an improved or well-founded version of the threshold density criterion proposed by de Boyer Montégut *et al*. (2004) to define the MLD. One of our results showed that MLD methodologies based on density thresholds of about 0.125 kgm-3 along the mixed layer produce overestimated MLDs and are inadequate to define a well-mixed layer in energetic terms. Our methodology serves to better support the threshold values of existing MLD methodologies. However, its main feature is that it represents a unique and complete methodology to estimate the MLD.

We rewrote WB as you suggested and conducted a detailed analysis of WB from the resulting expression (*Eq. 8* in the new version of the manuscript). The features you mentioned were included and properly described (*pages 5-7, lines 135-155*). The second effect you mentioned reflects the relationship between the density inhomogeneity and the associated WB along the vertical, but does not negatively affect the MLD estimation. In fact, this relationship (*Eq. 8*) was used to propose the procedure to calculate the MLD.

We agree with you that a unique WB equipotential should not determine the MLD globally year-round. A detailed analysis of the WB metric (*subsection 2.2 Defining the mixed layer*) and our new results (*Figs. 4 and 11*) confirmed your observation. Turbulence and its associated energy levels are spatially and temporally variable on a global scale; thus, the water column's stratification and vertical homogenization are not spatially uniform through the seasons.

**Comment 10:** L336: "It is also easy to implement numerically." Easy compared to what?

**Answer 10:** We are sorry for not being clear in that sentence. The numerical implementation of our MLD methodology only requires the potential density profile referred to 0 dbar, which is easily obtained from simple survey ocean data or numerical data. The script to compute the MLD is very short, and its formulae are not complex. In that regard, our methodology is easy to implement numerically. We rewrote this paragraph, hoping to be clearer (*page 25, lines 435-437)*.

---

## Author Comment (AC2)

**Review of the manuscript: https://doi.org/10.5194/egusphere-2024-4079**

**Title: The global ocean mixed layer depth derived from an energy approach**

**Reviewer #3**

We appreciate all your comments. After a careful revision that incorporated your comments and those of the other two reviewers, this version has greatly improved. We trust that you will find the revisions substantial and consider our study a valuable contribution to mixed-layer depth research. Below is a description of the major modifications made to the manuscript:

- The methodology (subsection 2.1) was extended by including details concerning the connection of the work done by buoyancy (WB) with the turbulent kinetic energy budget.
- We revised our definition of the mixed layer and how to calculate its depth (subsection 2.2. Defining the mixed layer). The mixed layer depth (MLD) was not calculated with a unique WB equipotential.
- Since we adjusted the MLD calculation, we rewrote the Results and Discussion sections entirely. The Conclusions and Abstract were also rewritten accordingly.
- The Supplement was rewritten to provide a detailed comparison of our MLD methodology with the common ones; we included global monthly climatologies of the MLD and the WB at the MLD.
- The dataset containing the monthly climatology of mixed layer depth and the derived variables calculated in this study is now publicly available at SEANOE (https://doi.org/10.17882/106181).

Responses to specific comments are shown below.

**Summary**

**Comment 1:** The mixed layer is an important oceanographic parameter, playing a significant role in understanding upper-ocean dynamics and air-sea interactions. Therefore, accurately calculating the mixed layer depth (MLD) is crucial. Over time, many researchers have proposed various methods to estimate the MLD, which can generally be categorized into three types: threshold methods, energy-based methods, and geometric shape methods. However, due to the very nature of the mixed layer, there is no perfect method. How can we quantitatively define "mixing"? It always requires some reference values, and the use of such reference values inherently leads to spatial or temporal dependence in the applicability of different methods.

This work approaches the problem from an energy perspective, proposing a new method for calculating MLD based on the amount of buoyancy work. The authors demonstrate through practical applications that this method performs reasonably well. Overall, I find this to be a very interesting study. It not only helps us better understand the mixed layer, but also provides an additional option for calculating its depth.

**Answer 1:** We really appreciate that you find our approach to estimating the MLD interesting. We agree with you that in MLD studies, it is necessary to clarify what aspects of the mixed layer are being studied and the considerations taken. That information was properly described in *subsection 2.2 Defining the mixed layer (page 5, lines 124-129)*.

**Comments**

**Comment 2:** I believe the authors may have overstated the performance of their proposed method. In the abstract, they claim that this method provides "a robust criterion based on physical principles." However, it should be noted that: 1) The authors still use a threshold value (20 J/m³) as the criterion for defining the mixed layer. This value is derived only from a few observational sections, and whether it is applicable globally remains questionable. 2) The authors' evaluation of "good" or "bad" performance seems to focus on regions where traditional methods do not perform well. Is this method better across all seasons and global ocean regions? In areas where density profiles change gradually, the MLD itself is inherently ambiguous—how should a "good" standard be defined in such cases? 3) The authors mention in the introduction that "in regions with vertically compensated layers, the density threshold may overestimate the MLD." Can the proposed method in this paper avoid this issue? Considering that the method is still based on density, it seems unlikely that this problem can be completely avoided.

**Answer 2:** Thanks for the comment. We acknowledge that in the prior version of the manuscript, we derived our results based on non-robust arguments. To fix that, we proposed a quantitative procedure to calculate the MLD; it is applicable in regions where density profiles change gradually. We agree that it is not obvious that a unique WB equipotential should determine the MLD globally year-round, which was obtained from Eq. 8 and represented in Fig. 2; please refer to the detailed analysis of the WB metric in *subsections 2.1 and 2.2*. Results of the WB threshold characterizing the MLD are shown in *Figs. 4 and 11*. Thus, the MLD was no longer calculated with the WB equipotential of 20 J/m³. The new results are completely different from those of the prior version of the manuscript due to the adjusted definition of the mixed layer.

We agree with you that local results can not support global conclusions. To circumvent this, we compared (qualitatively and quantitatively) our methodology with others on a global scale, considering different metrics (*subsection 3.2. MLD methodologies intercomparison*). We removed the limitation of focusing only on "challenging regions". We found that there is a lack of consistency among the methodologies in determining the MLD, but also that all the methodologies perform well under the oceanographic conditions for which they were built, according to the parameter being addressed. Since it is almost impossible to ensure that the true value of the MLD in any location and time is known, the accuracy of any methodology can not be determined, that is, the closeness of any MLD estimation to the true value. When comparing MLD methodologies, we can only evaluate their precision: the closeness between the different MLD estimates. Thus, we evaluated the precision of our methodology on a global scale and removed ad-hoc arguments from the description of our results.

For vertically compensated layers, our methodology may overestimate the MLD, in a similar way that the density threshold criterion of de Boyer Montégut et al. (2004); nonetheless, using WB, we can measure the degree of inhomogeneity of the water column associated with the compensated layer and investigate whether it is intense enough to suppress mixing. While our methodology may not provide a better or more meaningful MLD estimate than other methodologies, it does measure the water column inhomogeneity in terms of energy, a unique feature that other methodologies lack (*pages 21-23, lines 389-393*).

**Comment 3:** Although the authors express the buoyancy work integral in the form of Equation 3, in essence, this equation is the depth integration of density anomalies. This means the method still heavily depends on the choice of threshold value—especially in summer with shallow mixed layers when density increases slowly with depth. In such cases, different thresholds could lead to significantly different results. I recommend the authors conduct a deeper discussion and analysis of the limitations of their method, rather than focusing solely on its advantages.

**Answer 3:** We agree with you that our methodology is a threshold method. The procedure to determine the MLD in terms of WB is based on a threshold of the density inhomogeneity along the mixed layer (*subsection 2.2 Defining the mixed layer*). By considering the density vertically integrated, our methodology could represent an improved or well-founded version of the threshold density criterion proposed by de Boyer Montégut *et al*. (2004) to define the MLD. As you suggested, we carried out a deeper discussion and analysis of the limitations of our methodology. We conducted a detailed analysis of the WB metric and its physical significance (*subsection 2.1 An energy measure of the vertical homogeneity of the water column*) and evaluated its performance by comparing it with other methodologies (*subsection 3.2. MLD methodologies intercomparison*). The *Discussion section* extends the analysis of our MLD methodology and its downsides.

**Comment 4:** Although there is no universally accepted standard method for determining MLD, there is a parameter that can reflect the effectiveness of a given method to some extent—the Quality Index (https://doi.org/10.1029/2003JC002157). I suggest the authors consider using this parameter to evaluate the performance of their method.

**Answer 4:** We really appreciate your suggestion, which we implemented (*page 11, lines 255-267*). In the analysis of our MLD climatology, we adapted the Quality Index of Lorbacher et al. (2006) to WB and calculated it (*Fig. 5*). Results showed that, according to the quality index, our methodology performs very well in almost all the world ocean year-round: 96.72% of the world ocean has quality index values larger than 0.8 and in only 0.03% of the world ocean we have values less than 0.5. There is room for improvement in the use of the quality index in MLD studies; however, it was useful to evaluate the general performance of our methodology.

**The global ocean mixed layer depth derived from an energy approach**

Color code to track changes in the manuscript

: Text that was removed from the manuscript

Text in green: Text that was added to the manuscript

: Text that was replaced by the following text in blue

Text in blue: Text that replaced the previous

**Abstract**

The mixed layer depth (MLD) is critical for understanding ocean-atmosphere interactions and internal ocean dynamics. Traditional methods for determining the MLD , commonly relying on constant temperature and density thresholds, may not adequately address spatial and temporal variations in local oceanographic conditions, limiting their global consistency and applicability. ~~To address this, we propose an energy-based methodology that defines the MLD as the depth at which the work done by buoyancy (WB) reaches 20 Jm⁻³. This approach provides a robust, globally consistent, and easy-to-implement criterion grounded in physical principles. Our methodology captures the upper ocean's well-mixed layer in energetic terms, aligning with turbulent boundary layer dynamics while maintaining quasi-homogeneity in density and temperature for most of the global ocean. A global monthly MLD climatology derived from this method demonstrates its reliability across diverse oceanic conditions and its accuracy in regions and seasons where conventional methods struggle. This energy-based approach~~ An energy-based definition of the mixed layer could be a more physically consistent alternative to address this. We propose a physically derived and energy-based methodology that defines the mixed layer as the energetically homogeneous upper ocean layer in which water parcels can move with little or no buoyancy work. The threshold in buoyancy work determining the mixed layer globally throughout the year was carefully investigated. This approach provides a robust criterion that is globally and temporally consistent and easy to implement. An energy-based global monthly MLD climatology demonstrated the reliability of the methodology across diverse ocean conditions and its usefulness for seasonal to climate time scale studies, from regional to large spatial scales. Our methodology aligns with turbulent boundary layer dynamics while maintaining quasi-homogeneity in energy, density, and temperature year-round for most of the global ocean. This study advances the development of MLD energy-based methodologies that could offer significant potential for advancing the study of dynamic, and thermodynamic processes, including heat content and vertical exchanges.  Our methodology could also serve as a robust tool for validating ocean circulation models and to support intercomparison studies in initiatives such as the Ocean Model Intercomparison Project (OMIP) and the International Coupled Model Intercomparison Project (CMIP). Future research will explore its applicability to high-frequency processes and regional variability, further enhancing its utility for understanding and modeling oceanic phenomena.

**Short Summary (500 characters, non-technical)**

The surface mixed layer depth (MLD), where ocean properties are uniform, is key to  ocean dynamics and ocean-atmosphere interactions. We propose an alternative definition of the mixed layer as the layer in which water parcels can move with little or no work. This approach provides realistic mixed layer depth estimates across space and time under diverse ocean conditions. It has potential implications for improving our understanding of various ocean-atmosphere phenomena, including dynamic and thermodynamic ones.

[revised manuscript text omitted]

**Supplement**
The Supplement was rewritten to provide a detailed comparison of our MLD methodology with the
common ones; we included global monthly climatologies of the MLD and the WB at the MLD.

---

## Author Comment (AC3)

**Review of the manuscript: https://doi.org/10.5194/egusphere-2024-4079**

**Title: The global ocean mixed layer depth derived from an energy approach**

**Reviewer #2**

We appreciate all your comments. After a careful revision that incorporated your comments and those of the other two reviewers, this version has greatly improved. We trust that you will find the revisions substantial and consider our study a valuable contribution to mixed-layer depth research. Below is a description of the major modifications made to the manuscript:

- The methodology (subsection 2.1) was extended by including details concerning the connection of the work done by buoyancy (WB) with the turbulent kinetic energy budget.
- We revised our definition of the mixed layer and how to calculate its depth (subsection 2.2. Defining the mixed layer). The mixed layer depth (MLD) was not calculated with a unique WB equipotential.
- Since we adjusted the MLD calculation, we rewrote the Results and Discussion sections entirely. The Conclusions and Abstract were also rewritten accordingly.
- The Supplement was rewritten to provide a detailed comparison of our MLD methodology with the common ones; we included global monthly climatologies of the MLD and the WB at the MLD.
- The dataset containing the monthly climatology of mixed layer depth and the derived variables calculated in this study is now publicly available at SEANOE (https://doi.org/10.17882/106181).

Responses to specific comments are shown below.

**Summary**

**Comment 1:** Many methods have been proposed to identify the mixed-layer depth (MLD), which can be broadly classified into two categories, threshold method and energy method. This article is part of the overall effort to determine the MLD by using an energy based method (EBM). This choice is valuable because scalar threshold-based methods are subjective and not necessarily consistent with the physics of vertical mixing. This is not the case for the EMB method, because it is based on the work of the buoyancy force (WB) on the vertical.

**Answer 1:** We really appreciate that you find our approach to estimating the MLD significant.

**Major points**

**Comment 2:** Basics of this paper come from Reichl et al. (2022), who derive MLD from the concept of potential energy. However I have not clearly identified the links between your approach and the Reichl et al. (2022) one. Therefore I suggest to discuss the common points, differences, put into perspective, your approach with that of Reichl et al. (2022) in Section 2.1. It would be interesting to propose a mathematical development, which shows that the WB threshold is a potential energy barrier for water parcels. I do not see this correspondence in the article. In the same spirit, the stratification index (SI = R z 0 N 2zdz, where N 2 = −g ρ∂ρ∂z) could also have been used.

**Answer 2:** Thank you for the comment. Although WB and the PE anomaly of Reichl et al. (2022) share some similarities, our work is not based on Reichl's. In *subsection 2.1 An energy measure of the vertical homogeneity of the water column (pages 4-5, lines 93-113)*, we show an alternative expression of WB that allows us to appreciate its connection with the turbulent kinetic energy budget and, thus, with the physics of boundary layer turbulence. We did not provide further analysis of such a connection because that is beyond the scope of this paper and is proposed for future research.

WB represents an energy measure of the vertical homogeneity of the water column. In that respect, the WB values at different depths quantify the energy barriers for the vertical displacement of water parcels. Additional mathematical development (*Eqs. 4-8*) further describes the meaning and significance of WB. Comparing WB with other stratification indexes (including the buoyancy frequency) is very interesting; however, such an analysis is beyond the scope of this study and is proposed for future research.

**Comment 3:** The original contribution of this paper is to propose a WB threshold equal to $20 W.m^{-2}$ to define global MLDs. However this constant WB threshold was derived from 6 transects in the Pacific ocean and I am wondering why this threshold should be constant everywhere in space and time? I think this point needs to be argued because "local" conclusions; i.e. along transects; might not be valid at the global scale. For example, we see in Figure 4 that EBM does not always provide the "best" MLD estimates compared to other methods, knowing that "best" in this paper (see Section 3.2.2) is a visual criterion on the density profile, which is finally the classic scalar threshold. This last point shows that references have to be provided to define what is "best". Probably the definition of MLD depends on the question to be addressed. Is the EBM-derived MLD close to the density threshold-derived mixed-layer or to the turbulent kinetic energy threshold-derived mixing-layer (turbocline)?

**Answer 3:** Thank you for the comment. We agree with you that local results can not support global conclusions. To circumvent this, we compared (qualitatively and quantitatively) our methodology with others on a global scale, considering different metrics (*subsection 3.2. MLD methodologies intercomparison*). We removed the limitation of focusing only on "challenging regions". We found that there is a lack of consistency among the methodologies in determining the MLD, but also that all the methodologies perform well under the oceanographic conditions for which they were built, according to the parameter being addressed. Since it is almost impossible to ensure that the true value of the MLD in any location and time is known, the accuracy of any methodology can not be determined, that is, the closeness of any MLD estimation to the true value. When comparing MLD methodologies, we can only evaluate their precision: the closeness between the different MLD estimates. Thus, we evaluated the precision of our methodology on a global scale and removed ad-hoc arguments from the description of our results.

Our methodology is a threshold method. The procedure to determine the MLD in terms of WB is based on a threshold of the density inhomogeneity along the mixed layer (*subsection 2.2: page 5, lines 129-133 and pages 7-8, lines 156-184*). By considering the density vertically integrated, our methodology could represent an improved or well-founded version of the threshold density criterion proposed by de Boyer Montégut *et al*. (2004) to define the MLD. Thus, we could conclude that our methodology is closer to the density threshold than to the turbulent kinetic energy threshold derived from the turbulence theory.

**Minor points**

**Comment 4:** Line 82 : Replace the lower bound of the integral zref by zeq in Equation 3.

**Answer 4:** Done.

**Comment 5:** Line 95 : This sentence is confusing. We see in Figure 1 that all density variations are associated with an increase in WB. Please clarify.

**Answer 5:** Suggestion accepted (*page 5, lines 120-122*).
*"... the stratified profile (Fig. 1d) has a larger density variation than the winter profile (Fig. 1b), but the WB variation is larger in the winter than in the stratified profile: large density variations do not always correspond to large WB values."*

**Comment 6:** Line 146 : Describe in few words the methods of Holte and Talley (2009) and Romero et al. (2023).

**Answer 6:** Suggestion accepted (*page 9, lines 208-212*).
*"The third common MLD methodology is the multi-criteria method of Holte and Talley (2009), which calculates possible MLDs derived from threshold and gradient methods to select a final MLD estimate based on physical features in the profile. The recent MLD methodology is the sigmoid function fitting method of Romero et al. (2023), which computes the MLD and the maximum thermocline depth by evaluating the fit of the sigmoid function to the temperature profile."*

**Comment 7:** Lines 177-181 : Sorry but I do not understand the procedure to select the WB threshold. That is what I understand. For a given $W_{Bz}$ you obtain a family of $W_{Bi}$ with $i = 1, ..., n$ and each pair $(i, i + 1)$ corresponds to a depth variation $\Delta z_i$. The minimum $\Delta z_i$ selects $z_i$ and finally the $W_B$ threshold. Please clarify. In fact you could identify the structural change in $W_B$ just from $W_{Bz}$. Figure 2 shows smaller depth differences between $W_{Bz} = 1.5$ and $W_{Bz} = 2.5$ than between $W_B = 10$ and $W_B = 35$, suggesting that $W_{Bz} = 2$ might be a good threshold. So my questions are : Why is WB a better metric than WBz? Why do you need WBz to get a WB threshold? Please correct me and rewrite the lines 177-181 to clarify this important point.

**Answer 7:** We are sorry for the lack of clarity in that part of the procedure. However, in the revised version of the manuscript, we no longer use the first derivative of WB to find the MLD. We hope you find the new approach adequate (*subsection 2.2: page 5, lines 129-133 and pages 7-8, lines 156-184*).

**Comment 8:** Line 217 : Please give a reference for the inter-quartile formula, which detect outliers. Note that this formula is valid for a Normal distribution. Is it the case here?

**Answer 8:** You are right that the formula to detect outliers is valid for normal distributions. In the revised version of the manuscript, the intercomparison of different MLD methodologies was made on a global scale. We removed the limitation of focusing only on "challenging regions"; thus, the detection of outliers was no longer needed.

**Comment 9:** Line 250-254 : "EBM better identifies the relatively homogeneous upper ocean layer ..." This sentence is quite subjective, "best" is relative to the eye's estimation. For example on Figure 4c, I find HT09 better than EBM.

**Answer 9:** Please refer to answer 3 for a response to this comment.

**Comment 10:** Section 3.3 : It would be instructive to compare the EBM monthly MLD climatology shown Figure 5 with that of Boyer Montégut.

**Answer 10:** Thank you for your suggestion, which we implemented. In *subsection 3.2 MLD methodologies intercomparison*, we compared (qualitatively and quantitatively) our methodology with others on a global scale, considering different metrics. In the *Supplement*, we have maps of MLD climatologies obtained with different methodologies, which further support the comparison.

**Comment 11:** Line 275-279: MLDs around 600-700 m in the Greenland, Labrador, Iceland, Norway Seas, southern Pacific and Indian Oceans seem shallow. Another climatology would be helpful (Boyer Montégut?)

**Answer 11:** Thank you for your suggestion. In *subsection 3.2 MLD methodologies intercomparison*, we compared (qualitatively and quantitatively) our methodology with others on a global scale, considering different metrics. In the *Supplement*, we have maps of MLD climatologies obtained with different methodologies, which further support the comparison. The new MLD estimates in those regions are larger than the previous ones. The MLD can reach values of up to 945 m in the south of Iceland and the Labrador Sea, 1074 m in the Greenland, Iceland, and Norwegian seas region, and 614 m in the South Pacific and South Indian Oceans (*Page 11, lines 241-245)*.

**Comment 12:** Line 285-286 : Mention that the deepest MLDs are collocated with strong winds in these regions, add references.

**Answer 12:** Thanks for the comment. We are sorry for the omission. That sentence was removed in the new version of the manuscript. However, we are very careful not to omit relevant references throughout the manuscript.

**Comment 13:** Figure 6 : What is the unit of density in the lower panel ? I do not understand what is plotted, is it the density difference between the mixed-layer base and z = 10 m? Higher/lower densities occur in summer/winter. I would expect the opposite. But ok if it is the difference. Please check.

**Answer 13:** Of course. We referred to the histogram (expressed in density of probability). In the new version of the manuscript, we plotted cumulative density functions (CDFs).

*Figures 7 and 8* in the prior version of the manuscript are now *Figs. 6 and 7*. In them, we plotted the absolute differences in potential density and conservative temperature from the reference depth of 10 m to the MLD, respectively. In constructing the mixed layer definition, the density variations along the mixed layer throughout the year were established; the majority of the world ocean should have potential density differences in the interval (0.0150, 0.0300) kgm-3. The differences in conservative temperature from the reference depth of 10 m to the MLD shown in *Fig. 7* are heterogeneous in space and change over time with a type of seasonal variation. The temperature differences are generally large for large MLDs and vice versa; however, the temperature differences do not have the same structure or seasonal variation as those of the MLD. We found that the energy-based mixed layer is very close to quasi-homogeneity in density and temperature: almost 100% of the world ocean has density differences of less than 0.03 kg m-3, and 95% of the world ocean has temperature differences of less than 0.2°C throughout the year (*page 15, lines 275-285 and page 16, lines 286-293)*.

**Comment 14:** Line 288-289 : How do you explain the bimodal distribution?

**Answer 14:** Thank you for the comment; this is a very good observation. This bimodality is because of the region we used to compute the MLD histograms (*Fig. 6* in the prior version of the manuscript). In this region, we consistently have a region with large MLDs and a region to the north with consistently smaller MLDs. That structure in the MLD is the origin of the observed bimodality, which could be substantially reduced if we restrict the analysis to latitudes south of 50°S. In the new version of the manuscript, the Southern Ocean analysis is no longer present; we extended the analysis to a global scale, considering the different seasons (*subsection 3.2 MLD methodologies intercomparison*).

**Comment 15:** Line 291 : MLD variances are not shown, mention "not shown".

**Answer 15:** We are sorry for the omission. In the new version of the manuscript, the Southern Ocean analysis is no longer present; we extended the analysis to a global scale, considering the different seasons (*subsection 3.2 MLD methodologies intercomparison*).

**Comment 16:** Line 292-294 : The skewness of the distributions are not consistent with that of Johnson and Lyman (2022). Please explain why.

**Answer 16:** Thank you. This is a very good observation. The MLD should have persistence throughout the year; thus, the skewness of the MLD distribution is expected to remain with the same sign throughout the year. We have no insight into the result obtained by Johnson and Lyman (2022), but it would be interesting to investigate it. In the new version of the manuscript, the Southern Ocean analysis is no longer present; we extended the analysis to a global scale, considering the different seasons (*subsection 3.2 MLD methodologies intercomparison*). In the new version of the manuscript, we compared different MLD methodologies using different statistical metrics on a global scale during each season (*Table 1*). For each methodology, the skewness remains with the same sign throughout the year, which supports our prior claim.

**Comment 17:** Figure 7: Upper panel, another climatology would be useful for comparison. Lower panel, what is the variable and its unit on the "y" axis?

**Answer 17:** Thank you for your suggestion, which we implemented. In *subsection 3.2 MLD methodologies intercomparison*, we compared (qualitatively and quantitatively) our methodology with others on a global scale, considering different metrics. In the Supplement, we have maps of MLD climatologies obtained with different methodologies, which further support the comparison.

Regarding the units on the "*y*" axis, we referred to the histogram (expressed in density of probability). In the new version of the manuscript, we plotted cumulative density functions (CDFs).

**Comment 18:** Figure 8: Same remarks as Figure 7.

**Answer 18:** Thank you for your suggestion, which we implemented. In *subsection 3.2 MLD methodologies intercomparison,* we compared (qualitatively and quantitatively) our methodology with others on a global scale, considering different metrics. In the Supplement, we have maps of MLD climatologies obtained with different methodologies, which further support the comparison.

Regarding the units on the "*y*" axis, we referred to the histogram (expressed in density of probability). In the new version of the manuscript, we plotted cumulative density functions (CDFs).

**Comment 19:** Lines 397-400 : These density and temperature thresholds are not so far from the usual ones. Does that means that your EBM-derived MLD climatology does not depart significantly from other climatologies? As mentioned above (point 5) it would be instructive to compare the Figure 5 with another climatology.

**Answer 19:** Thank you for your suggestion, which we implemented. In *subsection 3.2 MLD methodologies intercomparison*, we compared (qualitatively and quantitatively) our methodology with others on a global scale, considering different metrics. In the Supplement, we have maps of MLD climatologies obtained with different methodologies, which further support the comparison. When comparing MLD methodologies, we evaluated the precision of our methodology, i.e, the closeness between the MLD estimates obtained with different methodologies. We found that our methodology is precise.

**The global ocean mixed layer depth derived from an energy approach**

Color code to track changes in the manuscript

: Text that was removed from the manuscript

Text in green: Text that was added to the manuscript

: Text that was replaced by the following text in blue

Text in blue: Text that replaced the previous

**Abstract**

The mixed layer depth (MLD) is critical for understanding ocean-atmosphere interactions and internal ocean dynamics. Traditional methods for determining the MLD , commonly relying on constant temperature and density thresholds, may not adequately address spatial and temporal variations in local oceanographic conditions, limiting their global consistency and applicability. ~~To address this, we propose an energy-based methodology that defines the MLD as the depth at which the work done by buoyancy (WB) reaches 20 Jm⁻³. This approach provides a robust, globally consistent, and easy-to-implement criterion grounded in physical principles. Our methodology captures the upper ocean's well-mixed layer in energetic terms, aligning with turbulent boundary layer dynamics while maintaining quasi-homogeneity in density and temperature for most of the global ocean. A global monthly MLD climatology derived from this method demonstrates its reliability across diverse oceanic conditions and its accuracy in regions and seasons where conventional methods struggle. This energy-based approach~~ An energy-based definition of the mixed layer could be a more physically consistent alternative to address this. We propose a physically derived and energy-based methodology that defines the mixed layer as the energetically homogeneous upper ocean layer in which water parcels can move with little or no buoyancy work. The threshold in buoyancy work determining the mixed layer globally throughout the year was carefully investigated. This approach provides a robust criterion that is globally and temporally consistent and easy to implement. An energy-based global monthly MLD climatology demonstrated the reliability of the methodology across diverse ocean conditions and its usefulness for seasonal to climate time scale studies, from regional to large spatial scales. Our methodology aligns with turbulent boundary layer dynamics while maintaining quasi-homogeneity in energy, density, and temperature year-round for most of the global ocean. This study advances the development of MLD energy-based methodologies that could offer significant potential for advancing the study of dynamic, and thermodynamic processes, including heat content and vertical exchanges.  Our methodology could also serve as a robust tool for validating ocean circulation models and to support intercomparison studies in initiatives such as the Ocean Model Intercomparison Project (OMIP) and the International Coupled Model Intercomparison Project (CMIP). Future research will explore its applicability to high-frequency processes and regional variability, further enhancing its utility for understanding and modeling oceanic phenomena.

**Short Summary (500 characters, non-technical)**

The surface mixed layer depth (MLD), where ocean properties are uniform, is key to  ocean dynamics and ocean-atmosphere interactions. We propose an alternative definition of the mixed layer as the layer
in which water parcels can move with little or no work. This approach provides realistic mixed layer
depth estimates across space and time under diverse ocean conditions. It has potential implications for
improving our understanding of various ocean-atmosphere phenomena, including dynamic and
thermodynamic ones.

[revised manuscript text omitted]

**Supplement**
The Supplement was rewritten to provide a detailed comparison of our MLD methodology with the
common ones; we included global monthly climatologies of the MLD and the WB at the MLD.

---

## Referee Report (RR1)

**Review of the paper entitled**
**The global ocean mixed layer depth derived from an energy approach**

I thank the authors for responding accurately to my questions and for extensively editing the manuscript. I realize that it was a huge but necessary effort to ultimately produce a much clearer and more convincing version. I suggest to the autors to turn off the "trackchange" mode to have a manuscript that is easier to read. In this final form, I accept the article for publication.

Lines 983-1002 : I appreciated this discussion. You mention that the EBM-MLD intrinsically depends on the $\Delta \overline{\rho^\theta}$ threshold, which may negatively influence its performance. I am wondering how to overcome this threshold. Following Equation 8, if you impose $WB = 0$ (or $WB$ small), then $\rho(h) = \overline{\rho}$. In that way, we can construct the following iterative process to obtain the MLD $h$ :

$$h^{n+1} = \eta - \frac{1}{\rho(h^n)} \int_{h^n}^{\eta} \rho(z)dz \qquad \text{where } n \text{ is the iteration} \tag{1}$$

$h$ is defined when $|h^{n+1} - h^n| \leq \epsilon$ where $\epsilon$ is your convergence criteria.

**1 Minor Points**

- Line 163 : Replace the lower bound of the integral $z_{ref}$ by $z_{eq}$ in Equation 3.

  Line 194 : Replace "the time integral of the buoyancy flux" by "the time integral of the surface buoyancy flux"

---

## Author Response (AR2)

**Review of the manuscript: https://doi.org/10.5194/egusphere-2024-4079**

**Title: The global ocean mixed layer depth derived from an energy approach**

**Reviewer #1 Brandon Reichl**

**Summary**

**Comment 1:** This is my second time reviewing this manuscript, which proposes an alternative definition of the ocean surface mixed layer depth (MLD) based on a buoyancy work diagnostic. I found the revision partially satisfied my previous concerns, but presented several new concerns detailed below.

**Answer 1:** We thank the reviewer for taking the time to evaluate our revised manuscript. In this revision, we strived to thoroughly address your questions and carefully consider all additional points raised in this round. We hope that you will find the revisions complete and consider our study a valuable contribution to the research on the mixed layer depth.

We prepared a marked-up manuscript version that shows the changes we made, as well as a version without tracked changes for easy visualization. The references to pages and lines in the comments below correspond to the manuscript without tracked changes.

**Comment 2:** As stated from the previous version, I find the subject matter interesting, and the alternative buoyancy work "WB" metric MLD definition provides a useful contextualization of differences between traditional simple threshold MLD methods and methods like potential energy (PE) anomaly. However, I think the presentation still needs to better characterize differences between threshold methods, WB, and PE anomaly. I found the paper addresses how WB differs from threshold methods (Section 2.2 is a nice addition in that regard). But it doesn't really clarify how the WB metric differs from PE anomaly (and how they offer different perspectives).

**Answer 2:** We agree with you that we need to better contextualize our methodology, EBM, in relation to the PE anomaly methodology. We added two paragraphs (in section *2. Methodology and data* and section *4. Discussion*) to address that. *Page 8, lines 190-198 and page 26, lines 464-475*.

"*The MLD definition differs between the potential energy (PE) anomaly methodology proposed by Reichl et al. (2022) and the methodology presented here. The PE anomaly diagnoses the MLD as the depth to which a given energy could homogenize a layer of seawater; the PE anomaly relates to the turbulent kinetic energy budget of the ocean surface boundary layer and serves as a good proxy for mixing, resulting in MLD estimates that are representative of active boundary layer turbulence. On the other hand, EBM determines the MLD by quantifying the vertical homogeneity of the water column in terms of the required WB to displace a water parcel vertically. WB may be associated with the turbulent kinetic energy budget of the upper ocean layer since it relates to the buoyancy flux required to mix that column during a specific time period; however, we did not conduct an in-depth analysis of the connection between WB and the turbulence approach to the mixed layer formation, which we propose for future research.*"

"*Reichl et al. (2022) introduced a framework for defining the MLD based on the PE anomaly, suggesting a threshold range of 10-25 J m⁻². While this range offers a valuable benchmark, determining the optimal thresholds under all seasonal and regional conditions remains an open question. Future research could*

*determine optimal values of the PE anomaly for global application and explore this approach in regional contexts by assessing the sensitivity of the MLD estimates to different thresholds of the PE anomaly. By comparison, with EBM, we were able to find the energy values (in terms of WB) that define the MLD globally during all seasons. Moreover, EBM provides additional information not supplied by the existing MLD methodologies; WB represents the potential energy barrier to the vertical displacement of water parcels, which could complement the analyses of oceanic processes occurring in intermediate vertical sections, commonly associated with interchanges of properties along the water column, such as the flux of particulate organic matter from the surface to sediments (Kirillin et al., 2012; Omand et al., 2020), the vertical content of chlorophyll (Carvalho et al., 2017; Briseño Avena et al., 2020), and entrainment in barrier layers (Katsura et al., 2022). The application of WB to analyze those processes is beyond the scope of this study and is proposed for future research.*"

**Comment 3:** I found that the tone of this paper implies that WB has provided a new superior metric compared to PE anomaly, threshold methods, and algorithm methods. I personally didn't find that to be objectively proven, and would suggest adjusting the narrative to more objectively present WB in the context of the other metrics. I therefore still suggest major revisions to the text, with more specific comments below.

**Answer 3:** We apologize if the tone of our original manuscript suggested that our methodology is unequivocally superior. We have modified the wording to present EBM more objectively, alongside other existing metrics. Below, we detail the specific changes made (*page 1, lines 12-14* and *page 25, lines 437-438*):

*"This study promotes the development of MLD energy-based methodologies that could offer significant potential for advancing the study of dynamic and thermodynamic processes, including heat content and vertical exchanges."*

*"In this study, we present a methodology for calculating the MLD, based on physical principles and energy considerations, which promotes the development of energy-based methodologies."*

We hope that these revisions clarify that WB is one of several viable metrics for estimating the MLD. Although it does provide the unique feature of measuring water-column inhomogeneity in energy terms, we are not asserting that it is categorically "better" in every context. We appreciate the reviewer's feedback and trust that the revised text now presents EBM in a fairer and more objective context.

**General comments**

**Comment 4:** As is already noted in the text, equation 1 is valid for a small perturbation with a locally referenced potential density. It is probably true that the error in using a surface reference pressure is usually small, but I did not find the citation provided in response to my previous comment was sufficient to justify why it is adequate to use surface referenced potential density for the purpose of computing buoyancy work over large vertical displacements. There is an error associated with this assumption, it would be prudent to quantify if/when that error matters for these calculations. I would guess that the deepest winter convective mixed layers, e.g., in deep water production regions such as the Labrador Sea, are situations where the error could be significant. (It may be useful to consider the approach we took in Reichl et al., 2022, where we evaluated the conceptually similar error associated with assuming that surface referenced potential density was the homogenized quantity for PE, rather than conservative temperature and absolute salinity.)

**Answer 4:** We thank the reviewer for pointing out this observation, which is relevant to our method. We reviewed the mathematical development to define WB. According to Vallis (2017), the expression for the buoyancy force (Eq. (1) in the new version of the manuscript) is valid for vertical displacements larger than infinitesimal ones. From that, we calculated WB as the line integral of the buoyancy force along a trajectory, typically from the MLD to the free surface. Indeed, recent studies have utilized the integral $\int_{\mathrm{MLD}}^{0} \left[ \rho^{\theta}(z) - \rho^{\theta}(\mathrm{MLD}) \right] dz$ to examine the MLD and mixing in the Indian and Southern Oceans (Lee et al., 2011; Small et al., 2021; Caneill et al., 2024). *Pages 3-4, lines 73-85*.

Regarding the error associated with using the potential density referred to 0 dbar, we addressed this when we explored the correspondence between WB and columnar buoyancy (see answer to Comment 5).

**Comment 5:** I spent much time trying to understand the arguments on the connection of WB to boundary layer turbulence that surrounds equations 4-7. I find the text to be vague with cursory arguments.

- Equation 4 (similar to eq 1) is only appropriate for infinitesimal vertical displacements with a locally referenced potential density. There is an error associated with the approximation applying it to large vertical displacements. It should be explained.
- The manipulation to go from equation 6 to equation 7 is not explained/justified.
- Equation 7 is dependent on the definition of the vertical coordinate "z", one could add an arbitrary constant "delta" to "z" and it changes the answer. Equation 3 is independent of the definition of the vertical coordinate, so this indicates that there is a problem with equation 7 and its interpretation.
- Equation 7 is compared to a 2008 paper by Herrmann et al., where a similar looking equation is presented. However, Herrmann et al. appear to consider bulk mixed layer dynamics in the derivation and cited work. Thus, their $N^2(h)$ is the buoyancy frequency evaluated at the mixed layer depth ("h") under a bulk mixed layer, and it is not the same as $N^2(z)$ in this paper. This difference is crucial because (a) by using "h" instead of "z" you should drop any dependence on the vertical coordinate, and (b) $N^2(h)$ in Herrmann et al. should be a function of time, the stratification at the MLD base changes as the bulk mixed layer deepens and is not equal to the stratification $N^2(z=h)$ in the initial condition (after some time/entrainment, the uniform density of the fluid within the mixed layer will be lighter due to mixing with the surface relative to the initial density at the depth of the MLD base, hence the density jump across the mixed layer base, $N^2(h)$ in Herrmann et al., is greater than $N^2(z=h)$ in equation 7. Maybe I missed something here (if so it could be explained better in the text), but for this reason I find it problematic to invoke the interpretation of equation 7 having the same meaning as Herrmann et al.'s equation. While the equations look similar, these distinctions seem to be critical for the connection to boundary layer turbulence energetics and, most importantly, mixing energy (that is most relevant for the TKE sink term) vs displacement energy.
- Equation 7 is interpreted to mean the "columnar buoyancy", or the buoyancy flux required to mix a water column to depth "h". I think what is important here about columnar buoyancy is that it is a buoyancy flux equivalent for the energy required to mix a water column over depth "h" (with some assumptions about TKE production and dissipation fraction). The relevant part here is not the columnar buoyancy itself, but that it yields an estimate of the energy requirement to homogenize the column.

**Answer 5:** Thank you for pointing out this observation; we agree with you that the explanation was unclear. We carefully reviewed the mathematical development to ensure the explanation is clear and supported by robust arguments.

Instead of using the delta of force in terms of $N^2$, we utilized an approximation derived by Caneill et al. (2024) to calculate columnar buoyancy in terms of the potential density referred to 0 dbar, which is valid for shallow depths. We examined the error associated with this approximation and found that it is small within the first few hundred meters, which is the focus of our study. Then, we showed that WB corresponds to the columnar buoyancy and thus, with the buoyancy flux required to mix that column during a given time period. Finally, using inductive reasoning, we suggested that it is plausible to connect WB with the turbulent kinetic energy budget of the upper ocean layer (Zippel et al., 2022) and, thus, with the physics of boundary layer turbulence. However, further analysis of this connection is beyond the scope of this study and is proposed for future research. *Page 4, lines 93-114*.

**Comment 6:** The text claims WB "advances the development of MLD energy-based methodologies", which might give the impression that the WB metric improves some inadequacy over the PE anomaly metric. I did not find a claim that WB advances any aspect of PE anomaly supported by the presented results. Another impression given by the text is that WB is aligned with boundary layer turbulence dynamics in a similar way to PE anomaly. I think the distinctions between WB and PE anomaly in that regard needs better clarified. While WB does use an energetically formulated criteria and it allows one to connect the threshold family of metrics to column integral aspects of the PE anomaly metric (this seems clear from section 2.2), it is otherwise distinct from PE anomaly in important ways (it measures displacement, rather than homogenization). This is not a criticism of the utility of WB, I think this concept nicely connects the threshold method to more fundamental fluid mechanics concepts via the work associated with particle displacement. But I wish to reemphasize a point from my previous review, that WB is not obviously aligned with turbulent boundary layer physics in the same way as PE anomaly, and thus does not obviously "advance" the PE anomaly method.

We discussed the energy requirement to homogenize a column of seawater (without a bulk mixed layer assumption) in detail in Reichl et al., 2022 (e.g., building to our equation 11, this is discussed by other work before ours, see references on PE anomaly within). If both energy-based approaches give the energetic distance from well-mixed (as indicated at L12, L57, L113, L131, L357, L417, L424, L432, etc.) then PE anomaly and WB should be equivalent metrics, but they are not. I therefore personally think PE anomaly is the energy metric more directly aligned with concepts mixing, at least these differences should be discussed in the text.

**Answer 6:** We apologize if the tone of our original manuscript suggested that our methodology advances the development of MLD energy-based methodologies. Our methodology contributes to the growth of those methodologies. We better contextualized our methodology (EBM) in relation to the PE anomaly methodology and added two paragraphs (in section *2. Methodology and data* and section *4. Discussion*) to address that. Please, see the answers to Comments 2 and 3.

**Comment 7:** I found section 2.2 a very useful addition to understand how WB and the threshold methods compare to one another. I found some of the emphasis on "energy" to be confusing though. "Energetically homogenous" is confusing to me, it sounds like the kinetic or potential energy of the column is homogenous? I think this is not the point, it is meant that a particle displaced in the vertical will not feel any net restoring forces along its path? This concept comes up throughout the text so it should be more clearly conveyed.

**Answer 7:** We apologize for the lack of clarity regarding what we mean by "energetically homogeneous". We reviewed and rewrote the first time the term "energetically homogeneous" appears, in the hope of explicitly clarifying the concept of "energetically homogeneous". *Page 5, lines 136-138.*

*"This study defines the mixed layer as the energetically homogeneous upper ocean layer; we consider a layer to be energetically homogeneous when water parcels can move with little or no WB within it.*

Additionally, we believe this concept is already described (albeit implicitly) at other points throughout the manuscript (*lines 6-7, 124-126, 149-151, 175-177, 178-188, 217-219, 268-270, 273-279, 289-290, 308-310, 315-318, 374-375, 483-484, and 546-547.*

**Specific comments**

**Comment 8:** L173: I didn't understand the justification for saying it works for "different regions and ocean conditions, such as polar seas, intermediate and deep water formation regions, and barrier and compensated layers", the remainder of the paragraph seems to indicate why it can struggle in certain scenarios? And makes the point why MLD computations that are globally applicable are difficult (a useful narrative!).

**Answer 8:** We agree that this phrase is confusing here because it was not yet justified; it was removed from this section and added in the conclusions (*page 28, lines 543-547*):

*"Based on energy considerations, we proposed an MLD methodology that is globally applicable and produces realistic estimates of the MLD. This MLD methodology performs robustly across different regions and ocean conditions, including polar seas, intermediate and deep-water formation regions, and barrier and compensated layer regions. The mixed layer, determined by energy processes, is quasi-homogeneous in energy, density, and temperature in most of the global ocean throughout the year."*

**Comment 9:** L205: Clarification: Does "ocean variables" mean that potential density, conservative temperature, and absolute salinity are each interpolated individually? So that it is not potential density computed from the interpolation of conservative temperature and absolute salinity?

**Answer 9:** The sentence was rewritten to be clearer (*page 9, lines 219-220*):

*"Potential density profiles were interpolated to 10 m if conservative temperature and absolute salinity measurements were not available at that depth; a linear interpolation of the potential density profile was implemented."*

**Comment 10:** L355-359: What is meant by physically realistic is unclear here. The threshold method still gives insight into the depth where the density changes by some value. It is a perfectly valid metric in that regard. It is important to clarify instead what the metrics measure relative to each other. I personally find the PE anomaly metric a better indicator of a mixing depth based on the arguments here than WB since the PE anomaly metric directly measures homogenization and WB measures displacement. Why is the concept of PE anomaly not discussed in this context?

**Answer 10:** You are right; the phrase "physically realistic" is confusing, and its use is not justified. Therefore, we removed the phrase "physically realistic" from the following sentences (*lines 373-374 and lines 395-396*), which now read as:

"*If the behavior of an MLD methodology deviates from the energy definition of the mixed layer, it is not energy-consistent.*"

"*The above analysis showed that B04D and EBM are energy-consistent, although WB in B04D is almost twice that of EBM in some regions and months, making it difficult to reconcile the large WB values of B04D with our mixed layer definition.*"

We agree with you that the threshold methodologies provide valid MLD estimations. We mention this in subsection *3.2. MLD methodologies intercomparison* (*page 20, lines 369-372*),

"*All the methodologies perform well under the oceanographic conditions for which they were designed, according to the parameter being addressed; Tang et al. (2025) evaluated 12 MLD methodologies and found that each has unique merits and limitations that depend on the analyzed ocean conditions. The determination of the best MLD methodology thus depends on the criterion used to rank the methodologies.*"

We better contextualized our methodology (EBM) in relation to the PE anomaly methodology, summarizing the characteristics of both methodologies and their differences; please, see the answers to Comments 2 and 3. Finally, it was not possible to include the PE anomaly methodology in subsection *3.2. MLD methodologies intercomparison* (*page 9, lines 222-224 and page 26, lines 466-468*).

"*It would have been very significant to consider the PE anomaly in the comparison; however, it was not possible due to the lack of a specific criterion to determine the MLD on a global scale.*"

"*Future research could determine optimal values of the PE anomaly for global application and explore this approach in regional contexts by assessing the sensitivity of the MLD estimates to different thresholds of the PE anomaly.*"

**Comment 11:** L435: Note that you could also compute PE anomaly from rho_0. These arguments about simple formulas and short scripts are entirely subjective. I personally don't think computing PE directly involved any complex formula, either. In fact, I would argue the calculation of rho_0 itself is much more complex than computing PE, particularly if using one of the high-order fits for the equation of state.

**Answer 11:** We fully agree with you that the PE anomaly can be computed from rho_0. However, since the metric used in our methodology is WB, we are confident that the statement written in lines 448-450 does not contradict the simplicity of its calculation or the merits of the PE anomaly. Thus, there is no need to mention other metrics in this sentence. Reilch et al. (2022) discuss the merits of the PE anomaly very well; readers can refer to their work for further information.

In the repository https://doi.org/10.5281/zenodo.1_4531829, we provide examples in Python, MATLAB, and R to illustrate the recipe for calculating WB, aiming to clarify this statement. As you clearly pointed out, calculating rho_0 could be very complex; however, that task is simplified by using the TEOS-10 toolbox.

**Comment 12:** L444: The energy required to homogenize a column of seawater is by definition where the PE anomaly concept comes from. If part of the utility of WB is as a "proxy" for this metric, why not use PE anomaly directly?

**Answer 12:** Of course, the PE anomaly diagnoses the MLD as the depth to which a given energy could homogenize a layer of seawater. The PE anomaly can be used to estimate the MLD. However, it is necessary to determine which energy value should be used to define the MLD globally across all seasons, which Reichl et al. (2022) did not provide (they suggested a threshold range of 10-25 J m$^{-2}$). They found that a spatially and temporally variable energy threshold should be used to reproduce, to some extent, MLDs similar to those obtained with the methodology of Holte and Talley (2009). *Page 26, lines 464-465 and page 27, 518-520.*

In addition, EBM could complement the existing MLD methodologies by providing additional information (*page 26, lines 468-475*):

"*By comparison, with EBM, we were able to find the energy values (in terms of WB) that define the MLD globally during all seasons. Moreover, EBM provides additional information not supplied by the existing MLD methodologies; WB represents the potential energy barrier to the vertical displacement of water parcels, which could complement the analyses of oceanic processes occurring in intermediate vertical sections, commonly associated with interchanges of properties along the water column, such as the flux of particulate organic matter from the surface to sediments (Kirillin et al., 2012; Omand et al., 2020), the vertical content of chlorophyll (Carvalho et al., 2017; Briseño Avena et al., 2020), and entrainment in barrier layers (Katsura et al., 2022). The application of WB to analyze those processes is beyond the scope of this study and is proposed for future research.*"

**Comment 13:** L450: The discussions surrounding PE anomaly (based on mixing rho_0) vs "delta PE" (based on mixing conservative temperature and absolute salinity) metrics in Reichl et al. (2022) seem extremely relevant to this conversation.

**Answer 13:** In the statement you mentioned, we are referring to commonly used MLD methods, not to energy-based methods. We properly included discussions regarding the EBM and PE anomaly methodologies in other more appropriate sections of the manuscript (see the answers to Comments 2, 6, and 7).

**Review of the manuscript: https://doi.org/10.5194/egusphere-2024-4079**

**Title: The global ocean mixed layer depth derived from an energy approach**

**Reviewer #2 Hervé Giordani**

**Summary**

**Comment 1:** I thank the authors for responding accurately to my questions and for extensively editing the manuscript. I realize that it was a huge but necessary effort to ultimately produce a much clearer and more convincing version. I suggest to the authors to turn off the "trackchange" mode to have a manuscript that is easier to read. In this final form, I accept the article for publication.

**Answer 1:** We thank the reviewer for taking the time to evaluate our revised manuscript and for finding our study a valuable contribution to the research on the mixed layer.

We have prepared a revised version of the manuscript, incorporating the comments of another reviewer. The changes in this version are the following:
1) We better contextualized our methodology, EBM, in relation to the PE anomaly methodology proposed by Reichl et al. (2022).
2) We reviewed the mathematical development to define WB.
3) We reviewed the mathematical development to explain the relationship between WB and columnar buoyancy, ensuring the explanation is clear and supported by robust arguments.

We prepared a marked-up manuscript version that shows the changes we made, as well as a version without tracked changes for easy visualization. The references to pages and lines in the comments below correspond to the manuscript without tracked changes.

**Comment 2:** Lines 983-1002: I appreciated this discussion. You mention that the EBM-MLD intrinsically depends on the $\Delta\rho^{\theta}$ threshold, which may negatively influence its performance. I am wondering how to overcome this threshold. Following Equation 8, if you impose WB = 0 (or WB small), then $\rho(h)=\rho$. In that way, we can construct the following iterative process to obtain the MLD $h$:

$$h^{n+1} = \eta - \frac{1}{\rho(h^n)} \int_{h^n}^{\eta} \rho(z)dz \qquad \text{where } n \text{ is the iteration} \qquad (1)$$

$h$ is defined when $|h^{n+1} - h^n| \le \epsilon$ where $\epsilon$ is your convergence criteria.

**Answer 2:** Thank you for your comment and suggestion to overcome the dependence on the $\Delta\rho^{\theta}$ value to define the MLD, which is very interesting. We will explore this proposal in future analyses to refine the MLD, which could potentially lead to revisiting the definition of the MLD.

**Minor points**

**Comment 3:** Line 163: Replace the lower bound of the integral $z_{\text{ref}}$ by $z_{\text{eq}}$ in Equation 3.

**Answer 3:** We apologize for that typo; we amended it. Please, see *Equation (2)*.

**Comment 4:** Line 194: Replace "the time integral of the buoyancy flux" by "the time integral of the surface buoyancy flux"

**Answer 4:** Thank you for your comment. We rewrite that sentence, which now reads (*page 4, lines 111-114*):

"*Columnar buoyancy represents the buoyancy loss required to mix the water column from the surface down to a depth of h during a specific time period (Lascaratos and Nittis, 1998; Herrmann et al., 2008); dividing columnar buoyancy by this time period gives the buoyancy flux required to mix that column during the same time period (Faure and Kawai, 2015).*"

---

## Author Response (AR3)

**Review of the manuscript: https://doi.org/10.5194/egusphere-2024-4079**

**Title: The global ocean mixed layer depth derived from an energy approach**

**Reviewer #1 Brandon Reichl**

**Summary**

The revisions satisfied many of my comments from the previous version. I have a few remaining comments, I would prefer they are addressed before recommending the manuscript for publication.

We thank the reviewer for taking the time to evaluate our revised manuscript; we are pleased that you found our response to be adequate. In this revised version, we have carefully considered all additional points raised in this round. We hope that you will find the revised manuscript complete and suitable for publication.

We prepared a marked-up manuscript version that shows the changes we made, as well as a version without tracked changes for easy visualization. The references to pages and lines in the comments below correspond to the manuscript with tracked changes.

**General comments**

**Comment 1:** More precise language could be used throughout the text where WB is referred to only as "energy", since there can be different energy approaches and considerations (e.g., potential energy anomaly). E.g., a more appropriate title could be "The global ocean mixed layer depth derived from buoyancy work".

**Answer 1:** We thank the reviewer for pointing out this observation. We agree with you; the title of the manuscript now reads as follows: "*The global ocean mixed layer depth derived from an energy approach based on buoyancy work.*" Additionally, in the main text of the manuscript, we emphasize that our MLD methodology is based on a specific energy approach, namely, the work done by buoyancy (*lines 193-195*).

**Comment 2:** I don't have a general issue stating that the details of the connection to boundary layer turbulence are beyond the scope of this work. However, that language is not always consistently reflected by the text. E.g., in the abstract it says this approach "aligns with turbulent boundary layer dynamics" yet later the claim is only that it is plausible to establish a relationship with the buoyancy flux, and hence to connect with the turbulent kinetic energy budget of the upper ocean. Since it is said the link between WB and turbulence boundary layer dynamics will be explored in future work I think it is more appropriate to leave it at that and to not overemphasize a connection that is only claimed as plausible in the text.

**Answer 2:** We agree that the wording can be occasionally confusing and inconsistent. We rewrote the sections related to this connection, both in the Abstract (*lines 15-18*) and the main text (*lines 121-133, 211-215, and 475-477*). Now, the wording is clear regarding the connection between WB and the turbulent approach to mixed layer formation, but it does not overemphasize a relationship that was not completely demonstrated.

**Comment 3:** I remain skeptical that this work establishes a precise, global and seasonally applicable range for WB. As I've mentioned throughout the review process, this is why we were reluctant to propose a global value of PE anomaly in Reichl et al. (2022). But ultimately a range of 12.5-20 J/m3 for WB is given in this work based on most of the ocean which has been profiled by Argo. This is ultimately not that different from the range of 10-25 J/m2 given as a plausible range in Reichl et al. 2022 (one I do not necessarily emphasize, I did not intend our work to come up with a PE anomaly value that reproduced Holte and Talley). So, (A) I personally don't think there is a significant difference between a range of 10-25 and 12.5-20 (yes it is a factor of 2, but neither range was formally derived). (B) Regardless of point A, the region where the values seem to differ the most from this mean value are the deep water formation regions. This is also where the quality index is lowest. This is one of the most important regions globally for interior water mass formation. I would have been interested for this study to comment more on the values of WB in those regions and its implications in the pursuit of a globally universal value. This might also emphasize regions where WB could provide additional insight beyond the delta threshold established in Section 3.1.

**Answer 3:** We conducted a detailed analysis of the precision of the MLD methodologies (see subsection 3.2. *MLD methodologies intercomparison*) and found that EBM is precise compared to other MLD methodologies. As described in subsections 3.1, EBM provides a realistic description of the MLD, consistent with the seasonal variation in ocean conditions across the ocean.

The WB values that determine the MLD are those shown in Fig. 4, not the 12.5-20 J m$^{-3}$ interval. The aim of subsection 3.3 is to very preliminarily explore whether a single WB value can determine the MLD on a global scale throughout the year; for this reason, we have analyzed only three meridional transects along three oceans. The analysis in subsection 3.3 was not intended to find a definitive WB range that determines the MLD across the global ocean throughout the year. The results of this subsection aim to motivate future research. We apologize for the lack of clarity in this regard; the title of subsection 3.3 was revised accordingly (*line 446*).

The sentence referring to the range of 10-25 J m$^{-2}$ in the PE anomaly found by Reichl et al. (2022) was removed from the Discussion section in lines 554-558. The sentence was rewritten to avoid confusion and placed in the paragraph where that study is discussed (*lines 499-502*).

As you suggested, we extended the analysis of the WB to provide more detailed comments on the values of WB in high-latitude and deep-water formation regions (*lines 300-311*), as well as the implications for pursuing a globally universal value (*lines 456-460*). In the Discussion section (*lines 563-576*), we had already described the limitations and scope of using the quality index in evaluating the MLD determination. This paragraph in the Discussion section outlines a roadmap to significantly enhance EBM by establishing a criterion to unequivocally determine the WB threshold that characterizes a well-mixed layer.

**Specific comments**

**Comment 4:** L47: I suggest to replace gravitational potential energy with potential energy anomaly..

**Answer 4:** Done (*line 53*).

**Comment 5:** Figure 3: Winter MLDs in convection regions approach 500-1000m by most MLD metrics (including WB as commented in the text). I suggest revisiting the colorbar to not saturate out the deepest values, a nonlinear/logarithmic increment may help. Similar for WB in Figure 4 and the saturated regions. These regions are geographically small, but have significant roles in interior ventilation.

**Answer 5:** Thank you for the pertinent suggestion. We agree with you that trying to show the spatiotemporal variability of the MLD (and that of WB) in a single map is very complex. There is great variation in the MLD and WB values across space and time that a single colorbar is not capable of describing in a proper way. To clearly appreciate and distinguish the characteristics of the MLD and WB, it is better to analyze specific regions (tropics, subtropics, or polar zones) during a specific season (summer or winter). A global map is a tool useful for a holistic view of the variables of interest.

For the above reasons, we provide the netCDF file of the MLD and associated variables (https://doi.org/10.17882/106181), allowing interested readers to plot their own maps for specific analyses. However, as you suggested, to emphasize the strong convective and deep-water formation regions, we present here maps of the MLD and the WB threshold, using nonlinear colorbars.

[Figure]

[Figure]

As is known, nonlinear (e.g., logarithmic) scales are non-intuitive, and their interpretation is difficult. We consider that, despite the limitations, the use of colorbars with linear scales is adequate to describe the spatiotemporal variability of the different variables throughout the manuscript. As mentioned, interested readers can plot their own maps using the provided database. Moreover, we are currently working on implementing a web-based data portal to display the EBM-MLD and its associated variables. We will include regional maps and linear and nonlinear colorbars to facilitate the analysis. This data portal will be released in due course; its URL will be made available in the source code repository (https://doi.org/10.5281/zenodo.14531829).

**Comment 6:** L112: "buoyancy losses required to mix" implies causation, but I think this is more of a statement of consistency and not causation? I suggest "buoyancy deficit compared to that of a homogeneous column in potential density". There seems to be some approximation here in the details, by assuming the potential density of the mixed column is the mean potential density of the unmixed column.

**Answer 6:** Thanks for the comment. You are right in that the definition of columnar buoyancy implies causation because the buoyancy loss must be provided to an initially stratified water column. However, we want to adhere to the original definition of columnar buoyancy provided by Lascaratos and Nitis (1998) and Herrmann et al. (2008). We consider that the details and approximations concerning columnar buoyancy and WB are adequately described in the manuscript. Please, see *lines 100-107*.

**Comment 7:** L123: I suggest clarifying it is "potential" density here.

**Answer 7:** Done (*line 136*).

**Comment 8:** L133: A connection to turbulence and boundary layers to me implies a connection to the mixing layer (by definition). Here WB is explicitly disconnected from the timescales of active mixing. Maybe I'm being too picky, but there seems to be some contradiction here (see also general comment "2" above).

**Answer 8:** Please, see the answer to Comment 2.

**Comment 9:** L169: WB_ref defined? I think I understand it, but I suggest writing the equation here to make it clear for the reader.

**Answer 9:** Done (*lines 182-183*).

**Comment 10:** S3.3: This ignores the deep convection regions, which happen to also be the regions where the quality index figure shows the worst performance (see also general comment "3" above).

**Answer 10:** Please, see the answer to Comment 3.